# Mechanisms underlying genome instability mediated by formation of foldback inversions in *Saccharomyces cerevisiae*

**Bin-zhong Li[1], Christopher D Putnam[1,2]\*, Richard David Kolodner[1,3,4,5]\***

[1]Ludwig Institute for Cancer Research, University of California School of Medicine, San Diego, San Diego, United States; [2]Departments of Medicine, University of California School of Medicine, San Diego, San Diego, United States; [3]Cellular and Molecular Medicine, University of California School of Medicine, San Diego, San Diego, United States; [4]Moores-UCSD Cancer Center, University of California School of Medicine, San Diego, San Diego, United States; [5]Institute of Genomic Medicine, University of California School of Medicine, San Diego, San Diego, United States

**Abstract** Foldback inversions, also called inverted duplications, have been observed in human genetic diseases and cancers. Here, we used a *Saccharomyces cerevisiae* genetic system that generates gross chromosomal rearrangements (GCRs) mediated by foldback inversions combined with whole-genome sequencing to study their formation. Foldback inversions were mediated by formation of single-stranded DNA hairpins. Two types of hairpins were identified: small-loop hairpins that were suppressed by *MRE11*, *SAE2*, *SLX1*, and *YKU80* and large-loop hairpins that were suppressed by *YEN1*, *TEL1*, *SWR1*, and *MRC1*. Analysis of CRISPR/Cas9-induced double strand breaks (DSBs) revealed that long-stem hairpin-forming sequences could form foldback inversions when proximal or distal to the DSB, whereas short-stem hairpin-forming sequences formed foldback inversions when proximal to the DSB. Finally, we found that foldback inversion GCRs were stabilized by secondary rearrangements, mostly mediated by different homologous recombination mechanisms including single-strand annealing; however, *POL32*-dependent break-induced replication did not appear to be involved forming secondary rearrangements.

**\*For correspondence:**
cdputnam@ucsd.edu (CDP);
rkolodner@health.ucsd.edu (RDK)

**Competing interests:** The authors declare that no competing interests exist.

## Introduction

Organisms invest substantial effort into properly replicating their genomes and preventing DNA damage from leading to the accumulation of mutations and gross chromosomal rearrangements (GCRs), such as translocations, deletions and inversions. Characteristic GCRs underlie many human diseases, including leukemias and lymphomas (*Mitelman et al., 2007*). In addition, ongoing accumulation of mutations and GCRs is seen in many types of cancer (*Campbell et al., 2010*; *Gibson et al., 2016*; *Gundem et al., 2015*; *Nowell, 1976*; *Uchi et al., 2016*). For many years, our understanding of the origin of GCRs in mammalian cells has been limited. Considerable recent progress has been made through the discovery of cancer susceptibility syndromes associated with increased rates of accumulation of mutations and GCRs as well as the use of whole genome sequencing (WGS) to characterize cancer genomes and to identify genome instability signatures such as chromothripsis and chromoplexy (*Baca et al., 2013*; *Li et al., 2020*; *Meyerson and Pellman, 2011*). In spite of this progress, a comprehensive understanding of the pathways and mechanisms that suppress or promote the formation of GCRs in mammalian cells is not yet available.

Using quantitative genetic assays for the formation of GCRs in the yeast *Saccharomyces cerevisiae*, we and others have used both hypothesis-based analysis of candidate genes and systematic genetic screens to identify genes and pathways that suppress the formation of GCRs (reviewed in *Putnam and Kolodner, 2017*). Many of the genes identified in these studies can be characterized as having important roles in (1) preventing the accumulation of DNA damage (e.g. *RAD27*, which encodes a flap endonuclease involved in Okazaki fragment maturation [*Kao and Bambara, 2003*]), (2) promoting the repair of DNA damage through conservative repair mechanisms, such as allelic homologous recombination (HR) (e.g. *RAD52*, which encodes a protein involved in loading the Rad51 strand-exchange protein [*Krogh and Symington, 2004*]), or (3) preventing the aberrant repair of DNA damage, which can act on intermediates that are normally processed by conservative repair pathways (e.g. *PIF1*, which encodes a DNA helicase that suppresses de novo telomere addition to double-strand breaks (DSBs) [*Myung et al., 2001*; *Schulz and Zakian, 1994*]).

Chromosomal translocations containing segments of DNA in inverted orientation, variously called foldback inversions, isoduplication translocations, and inverted duplications, have been observed in a number of studies on the formation of GCRs (*Pennaneach and Kolodner, 2004*; *Pennaneach and Kolodner, 2009*; *Putnam et al., 2014*). Inversions have also been observed in GCRs mediated by HR between inverted segments of DNA on the same chromosome and mediated by fragile sites and G-quadruplex forming sequences (*Nene et al., 2018*; *Schmidt et al., 2006*; *Srivatsan et al., 2018b*; *Zhang et al., 2013*). These types of inversions have been observed in human cancers, including pancreatic, ovarian, breast and esophageal cancer, are associated with poor prognosis in high-grade serous ovarian cancer, and occur in syndromes underlying autism, birth defects, developmental delay, and intellectual disability (*Ballif et al., 2003*; *Bonaglia et al., 2009*; *Campbell et al., 2010*; *Ford and Fried, 1986*; *Guenthoer et al., 2012*; *Hermetz et al., 2014*; *Stephens et al., 2009*; *Tanaka et al., 2007*; *Wang et al., 2017b*; *Yu and Graf, 2010*). Interestingly, some engineered palindromic inversions are unstable, are cleaved by the Mre11-Rad50-Xrs2 complex and Sae2 and induce GCRs (*Lobachev et al., 2002*).

Early studies of GCRs with inversion junctions provided an imperfect understanding of their structure, because the structures of these GCRs were generally not completely determined at nucleotide sequence resolution (*Pennaneach and Kolodner, 2004*). However, recent studies have determined the sequence of a limited number of inversion junctions, identified inverted sequences that mediate the formation of these inversion translocation junctions, and implicated the Mre11-Rad50-Xrs2 complex, Sae2, and Tel1 in preventing foldback inversions (*Deng et al., 2015*; *Liang et al., 2018*; *Putnam et al., 2014*). In the present study, we have systematically used WGS in combination with GCR assays to perform a detailed mechanistic analysis of the formation and suppression of foldback inversions. Our results establish the formation of single-stranded DNA (ssDNA) hairpins as key intermediates that channel DNA damage away from conservative repair mechanisms and toward the formation of GCRs, identify two mechanistically distinct types of hairpins, identify distinct pathways that prevent each type of hairpin from forming foldback inversions, and identify mechanistic steps involved in the conversion of hairpin intermediates into foldback inversions.

## Results

### *SAE2* and *MRE11* suppress foldback inversion GCRs

We first determined the effect of a *sae2Δ* deletion mutation, the *sae2-S267A* mutation, which eliminates an essential Cdk1-phosphorylation site on the Sae2 protein (*Huertas et al., 2008*), and the *sae2-MT9* mutation, which eliminates 9 Mec1/Tel1 phosphorylation sites on the Sae2 protein (*Baroni et al., 2004*), on the rates of accumulating GCRs selected using the unique-sequence-mediated GCR (uGCR) assay (*Figure 1A*; *Putnam et al., 2009*) present on the left arm of chromosome V (chrV L). Consistent with previous reports that the Sae2-S267A and Sae2-MT9 proteins are defective for Sae2 function (*Baroni et al., 2004*; *Huertas et al., 2008*), the *sae2Δ*, *sae2-S267A*, and *sae2-MT9* mutations caused equivalent increases in the uGCR rate. However, these GCR rates were lower than those caused by *mre11Δ* mutation and *mre11-H125N* mutation, the latter of which eliminates the Mre11 nuclease activity (*Figure 1B*; *Supplementary file 1*; *Liang et al., 2018*; *Putnam et al., 2009*).

We next analyzed GCR-containing strains by paired-end WGS (*Figure 1D,E*; *Supplementary files 2–6*; *Figure 1—figure supplement 1* and *Figure 1—source data 1*). In almost all cases (*sae2Δ*: 19

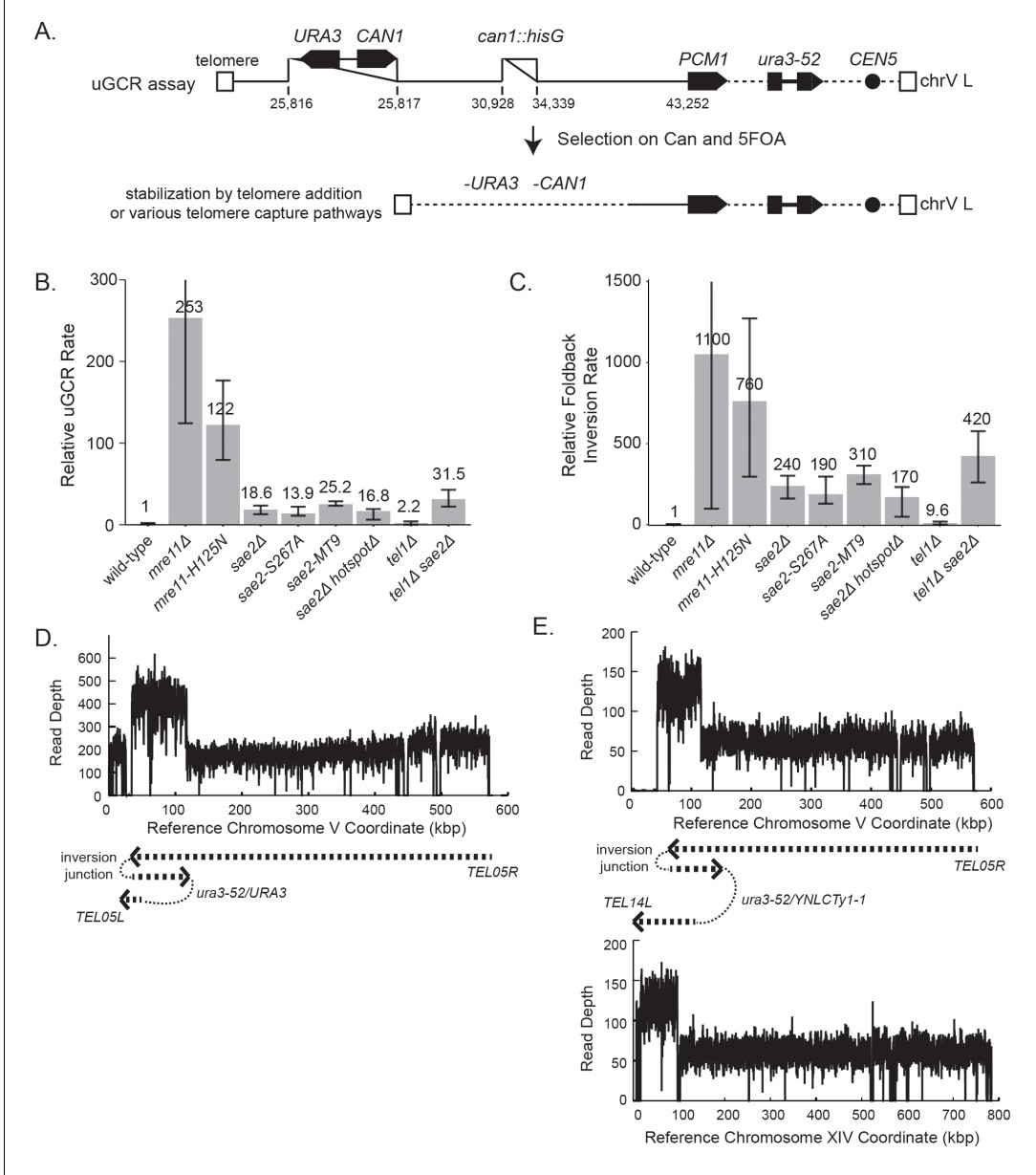

**Figure 1.** GCRs recovered in uGCR strains with *sae2* defects are primarily hairpin-mediated foldback inversions. (**A**) Diagram of the left arm of chromosome V (chrV L) containing the uGCR assay. The normal *CAN1* locus is deleted by substitution with a *hisG* fragment and a cassette containing the *CAN1* and *URA3* genes has been inserted into the *YEL068C* gene. Coordinates for the modified chrV L are reported as the reference genome coordinates for unmodified regions or reported as positions relative to the centromeric coordinate of the insertion site for inserted elements, for example chrV:34,339–110 is a position in the inserted *hisG* fragment 110 bases telomeric to the insertion site at chrV 34,339. Simultaneous selection against *CAN1* and *URA3* by canavanine (Can) and 5-fluoroorotic acid (5FOA) selects for GCRs that ultimately lose both *CAN1* and *URA3* and are stabilized by addition or capture of a telomere. (**B**) The relative uGCR rate for strains with *mre11* or *sae2* defects are displayed with error bars corresponding to the 95% confidence intervals. GCR rates are reported in ***Supplementary file 1***. (**C**) The relative foldback inversion GCR rate for strains with *mre11* or *sae2* defects are displayed as in panel B. (**D**) Example read depth plot determined by WGS for chrV from a foldback inversion GCR resolved by the *ura3-53/URA3* homology-mediated rearrangement. Thick-hashed arrows underneath the plot indicate the connectivity between the portions of the GCR that map to the different regions of the reference chromosome. (**E**) Example read depth plot for chrV and chrXIV from a foldback inversion GCR resolved by a *ura3-52/YNLCTy1-1* homology-mediated rearrangement displayed as in panel D, showing the duplication of the region of chrXIV between *TEL14L* and *YNLCTy1-1*.

The online version of this article includes the following source data and figure supplement(s) for figure 1:

**Source data 1.** Complex GCR structures.

**Figure supplement 1.** Example copy number plots for simple GCR structures not involving foldback inversions.

of 20 GCRs. *sae2-S267A*: 12 of 12 GCRs. *sae2-MT9*: 11 of 12 GCRs), WGS revealed features of a foldback inversion GCR on chrV L (*Figure 1D,E*; *Supplementary file 3*): (1) a deletion that spanned some or all of the counter-selected *CAN1/URA3* cassette; (2) a duplication that extended from the deletion to a more centromeric site; (3) discordant read pairs that indicated a novel inversion junction at the telomeric side of the duplication; and (4) reads that sequenced this novel inversion junction. Importantly, the resulting GCRs were not dicentric; these GCRs all underwent additional secondary rearrangements, such as homology-mediated rearrangements between the Ty retrotransposon-containing *ura3-52* and the *URA3* in the *CAN1/URA3* cassette (*Figure 1D*) and between *ura3-52* and *YNLCTy1-1* (*Figure 1E*). The secondary rearrangements will be described in detail below.

Using the GCR rates and GCR spectra obtained, we found that *sae2* mutations caused increases in the rate of accumulating foldback inversions that were ~240 fold higher than that of the wild-type strain (*Figure 1C*). In contrast, *mre11* mutations caused higher rates of accumulating foldback inversions than *sae2* mutations; however, this difference was less than the difference in total GCR rates observed for *mre11* and *sae2* mutants (*Figure 1B*). Mutations in *SAE2* almost exclusively caused the formation of foldback inversions (92–100% of the GCRs), whereas *mre11* mutants accumulated a diversity of GCRs including foldback inversions (30–50% of the GCRs; *Supplementary file 1*; *Chen and Kolodner, 1999*; *Liang et al., 2018*).

## Inversion junction sequences suggest a ssDNA hairpin intermediate

The sequences of virtually all of the inversion junctions in GCRs selected in *sae2* and *mre11* mutants, as well as the *sae2Δ* double mutant strains described below, suggested the involvement of ssDNA hairpin intermediates (*Figure 2*, *Figure 2—source data 1*). A mechanism for forming a foldback inversion consistent with this intermediate involves: (1) 5'-resection from a DSB or other initiating damage, exposing inverted ssDNA sequences; (2) formation of a ssDNA hairpin intermediate, which is a known substrate for the Sae2-Mre11-Rad50-Xrs2 endonuclease (*Oh and Symington, 2018*); (3) processing of any flap structures formed; and, (4) extension of the 3' terminus of the ssDNA hairpin by DNA synthesis (*Figure 2*). In two interesting cases, foldback inversions isolated from *sae2Δ* and *sae2Δ sgs1Δ yku80Δ* mutant strains had junction sequences consistent with a mechanism involving the formation of two sequential ssDNA hairpins in which the second hairpin was formed using sequences generated by extension of the first hairpin (*Figure 2—figure supplements 1,2*).

## Identification of an inversion hotspot sequence

Most foldback inversion GCRs isolated in *mre11-H125N*, *sae2Δ*, *sae2-S267A*, and *sae2-MT9* single mutant strains, as well as the *sae2Δ* double mutant strains described below, were mediated by an inversion hotspot sequence in the *can1::hisG* disruption cassette within the GCR assay breakpoint region (coordinates chrV 34,339–107_34,339–75; *Figure 3A*; *Supplementary file 3*). This hotspot, when present in ssDNA, is predicted to form a ssDNA hairpin with a 3 nt loop and a 15 bp stem (*Figure 3B*), which is the longest stem available for ssDNA hairpins in the uGCR assay breakpoint region (*Figure 3C*). In addition to the hotspot-mediated GCRs, we identified foldback inversions that were mediated by 62 other ssDNA hairpin-forming sequences present in the breakpoint region; however, these hairpin-forming sequences resulted in shorter stem structures ranging from 4 to 14 bp and were used less frequently (*Figure 3A,C*).

## The inversion hotspot does not induce the formation of GCRs

To test the hypothesis that the hotspot site promotes the formation of foldback inversions by causing DNA damage, we deleted the hotspot sequence in a *sae2Δ* uGCR assay-containing strain. The *sae2Δ hotspotΔ* strain had an overall uGCR rate and a foldback inversion GCR rate (9 of 12 GCRs were foldback inversions) that were similar to those of the *sae2Δ* strain containing the hotspot (*Figure 1B,C*). None of the inversion sites used in GCRs isolated from the *sae2Δ hotspotΔ* strain were next to the deleted hotspot or were used more than once (*Figure 3A*). These results argue against a role of the hotspot in inducing DNA damage, but would be consistent with a role of the inversion hotspot in redirecting DNA damage present in a *sae2Δ* strain into the formation of foldback inversion GCRs mediated by the inversion hotspot.

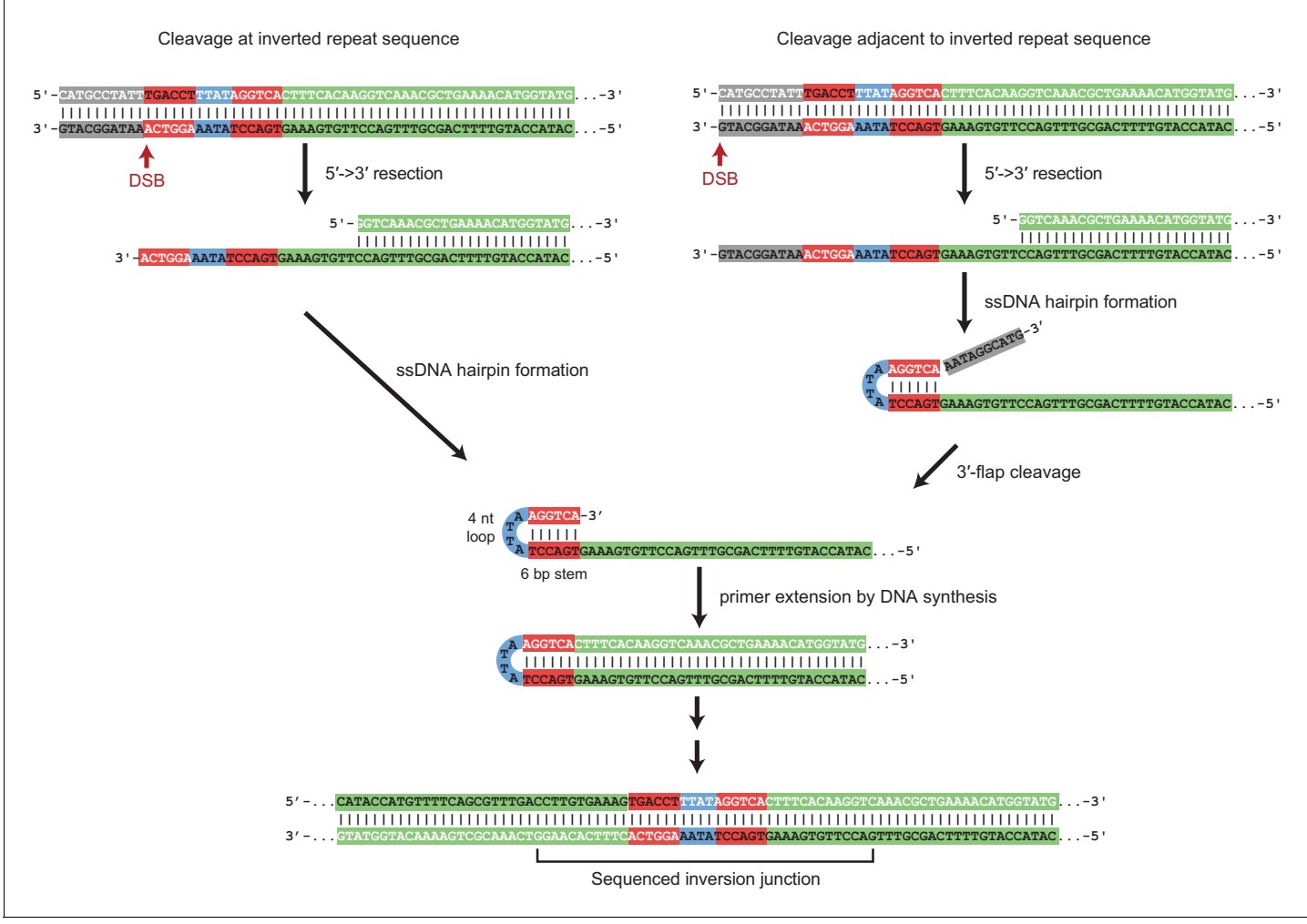

**Figure 2.** Foldback inversion GCRs are mediated by a ssDNA hairpin intermediate. Proposed mechanism underlying foldback inversion formation based on the inversion junction sequences recovered. 5′ to 3′ resection from a DSB or other initiating form of DNA damage exposes a ssDNA region predicted to form a ssDNA hairpin. Appropriately positioned DSBs can lead to properly paired hairpin stems, whereas other DSBs will lead to 3′ flaps that require processing before extension by DNA polymerases. Extension of the hairpin leads to the inversion junction sequences observed. The online version of this article includes the following source data and figure supplement(s) for figure 2:

**Source data 1.** Predicted ssDNA hairpin structures for observed foldback inversions.
**Figure supplement 1.** Inversion junction formed by two sequential ssDNA hairpin intermediates.
**Figure supplement 2.** Inversion junction formed by two sequential ssDNA hairpin intermediates.

## The inversion hotspot directs damage processing

To investigate the mechanism by which the hotspot sequence channels the processing of DNA damage into the formation of hotspot-mediated foldback inversions and whether the stability of the ssDNA hairpin at the hotspot sequence plays a role in this process, we induced defined DSBs on either side of the hotspot using CRISPR/Cas9, isolated GCR-containing strains, and analyzed them by WGS (*Figure 4*; *Supplementary file 3*). Induction of the centromeric DSB (chrV:34,470) places the hotspot sequence on the telomeric fragment where it cannot direct hairpin-mediated inversions oriented towards the centromere after resection. Consistent with this, the GCRs recovered from both the wild-type and *sae2Δ* uGCR strains did not involve the hotspot and were predominantly de novo telomere addition-mediated GCRs (11 of 24 GCRs) or microhomology-mediated translocations (11 of 24 GCRs) (*Figure 4*; *Supplementary file 3*). In contrast, cleavage at the telomeric DSB site (chrV:30,843) would allow resection and hotspot hairpin formation to generate centromere-oriented inversions. The GCRs recovered from the wild-type parental strain were predominantly de novo

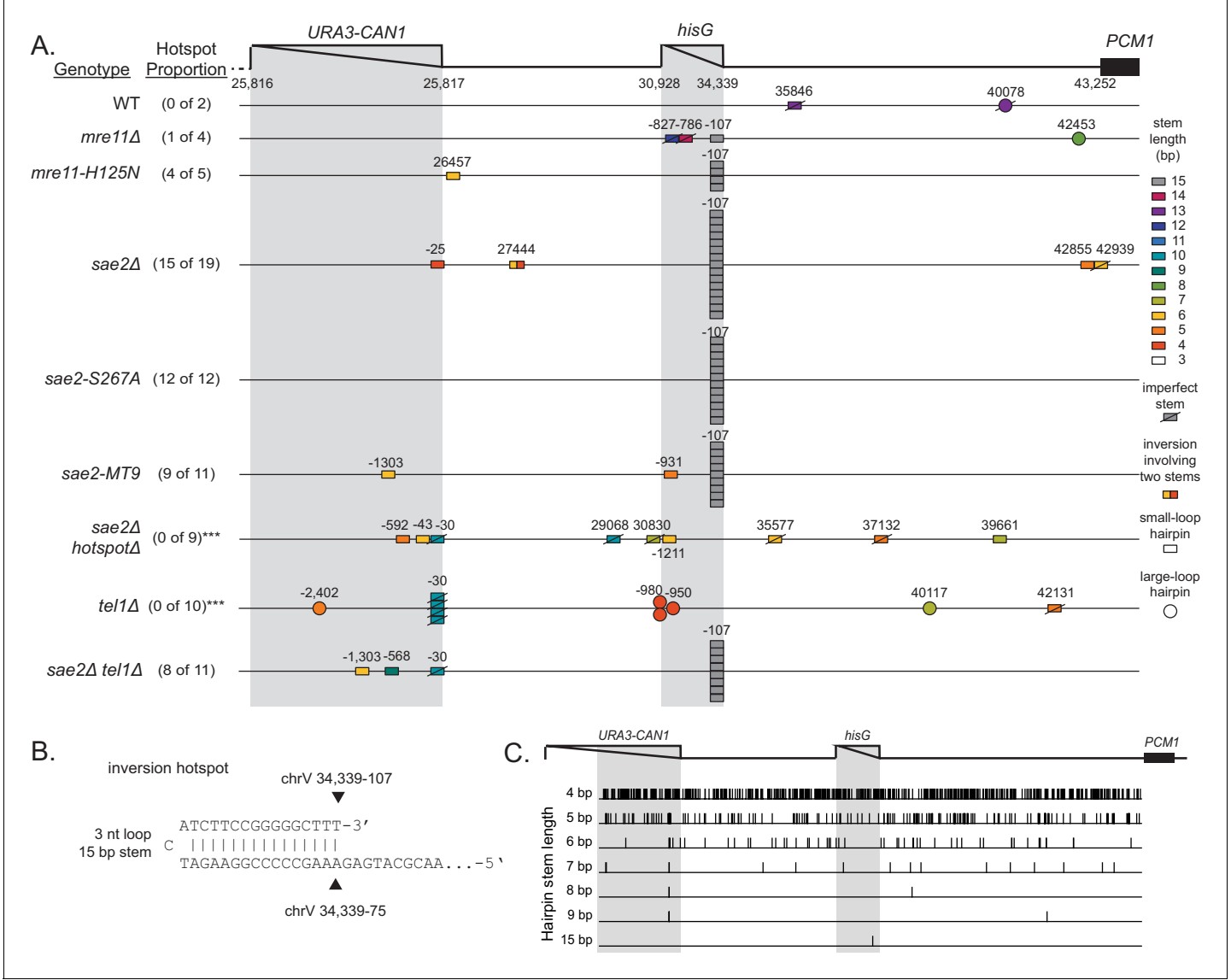

**Figure 3.** Distribution of inversion junctions in *mre11* and *sae2* mutants. (**A**) Map of the position of the inversion junctions (boxes) reveals that an inversion hotspot in the *hisG* insertion mediates a large proportion of the foldback inversions in strains with *mre11* and *sae2* defects. Each observed inversion is represented by a separate box (short loop hairpin) or separate circle (large loop hairpin) at the inversion junction position; for example, the 15 grey boxes corresponding to the inversion hotspot (labeled '−107') on the *sae2Δ* line correspond to 15 independent GCR-containing strains isolated from the *sae2Δ* single mutant that have an inversion at the hotspot. The only mutants whose usage of the inversion hotspot relative to other inversion sites was altered relative to the *sae2Δ* single mutant control were the *sae2Δ hotspotΔ* and *tel1Δ* mutants (p=0.0002 and p=5×10⁻⁵; Fisher's exact test). 'Imperfect stem' indicates stems predicted to contain one or more mispairs or unpaired bases by MFOLD (*Zuker, 2003*). (**B**) The predicted ssDNA hairpin for the inversion hotspot is predicted by MFOLD to form a 15 bp stem with a three nt loop. (**C**) The inversion hotspot contains the longest stem structure for any of the theoretically predicted sites with a propensity to form ssDNA hairpins in the uGCR chrV L breakpoint region. Predicted hairpin sites were restricted to a loop size of <50 nt and were not allowed to have mismatches in the stems.

telomere addition-mediated GCRs (11 of 12 GCRs), whereas the GCRs recovered from the *sae2Δ* mutant were primarily foldback inversions (10 of 12 GCRs; *Figure 4*; *Supplementary file 3*). Remarkably, only two of the foldback inversions recovered from the *sae2Δ* strain in this case were mediated by the hotspot, and eight were mediated by another hairpin-forming sequence immediately adjacent to the cleavage site (coordinates chrV:30,857_30,872; predicted hairpin with a 6 bp stem) (*Figure 4*; *Figure 2—source data 1*; *Supplementary file 3*).

We found the spectrum of GCRs induced by DSBs in the *sae2Δ* strain surprising. The centromeric cleavage (chrV:34,470) did not result in foldback inversions, and the telomeric cleavage

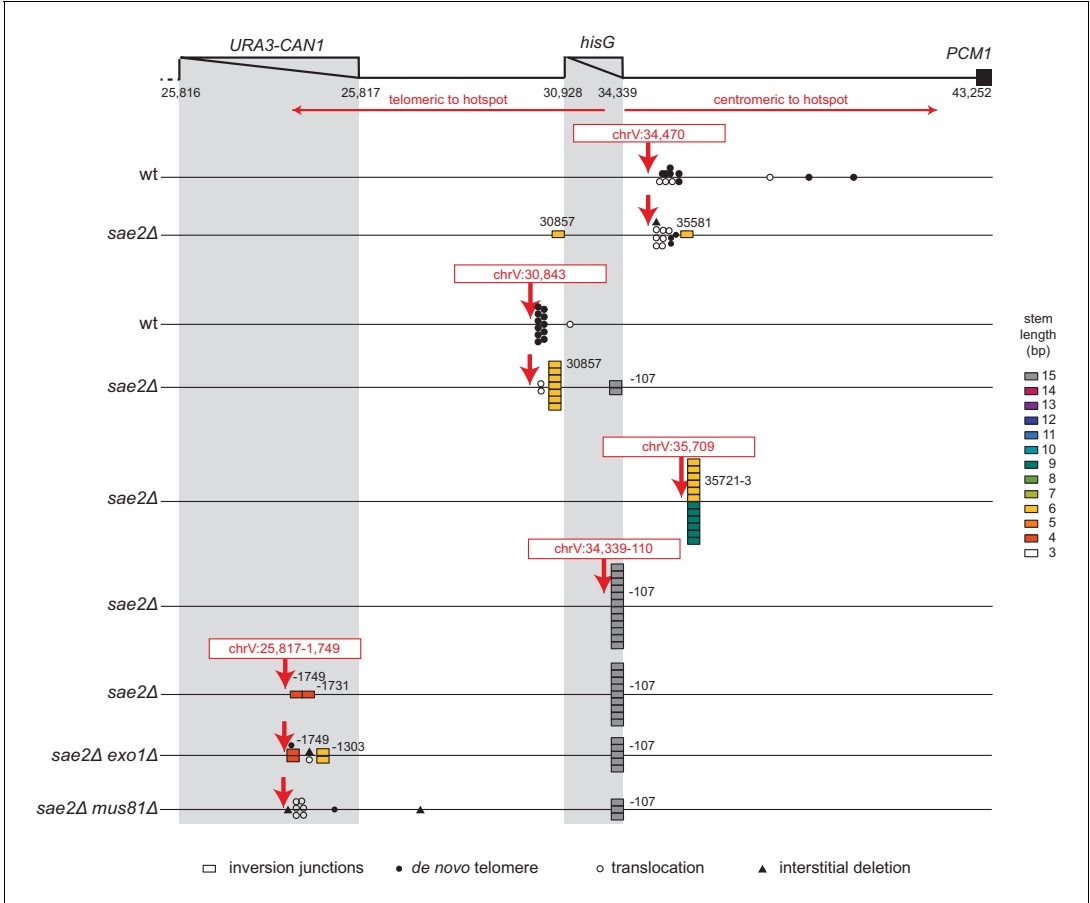

**Figure 4.** Analysis of GCRs generated by induction of site-specific DSBs. The position of CRISPR/Cas9 cleavage sites are indicated by red arrows. The sites at which various rearrangements were induced are indicated by different symbols: inversion junctions are shown as boxes, de novo telomere additions as filled circles, microhomology-mediated translocations as open circles, and microhomology-mediated interstitial deletions as filled triangles. The relevant genotype of the strains where the DSBs were induced is indicated on the left side.

The online version of this article includes the following figure supplement(s) for figure 4:

**Figure supplement 1.** Distribution of potential hairpin-forming sites centromeric to the induced DSBs.

(chrV:30,843) resulted in few hotspot-mediated foldback inversions. These effects could be due to the position of the induced DSB relative to hairpin-forming sequences: chrV:34,470 is centromeric to the hotspot and not adjacent to a stable hairpin-forming sequence, whereas chrV:30,843 is ~1 kb upstream from the hotspot and immediately adjacent to a hairpin-forming sequence.

We therefore tested three additional cleavage sites in the *sae2Δ* uGCR strain (*Figure 4*). The first cleavage site was at chrV:35,709, which was centromeric to the hotspot and adjacent to a previously observed inversion forming site; the GCRs recovered in this case were predominantly foldback inversions mediated by 1 of 2 adjacent hairpin-forming sequences capable of forming hairpin stems with lengths of 9 bp (6 of 12 GCRs) and 6 bp (6 of 12 GCRs) (*Figure 4*; *Supplementary file 3*). The second cleavage site was at chrV:25,817–1,754, which was within *CAN1*, >5 kb telomeric to the hotspot, and distal to any previously observed inversion-forming sites; in this case, the GCRs recovered were predominantly foldback inversions mediated by the hotspot sequence (9 of 11 GCRs), despite the fact that the chrV:25,817–1,754 cleavage site was ~4 kb further from the hotspot than the chrV:30,843 cleavage site (*Figure 4*; *Supplementary file 3*). The third cleavage site was at chrV:34,339–110, which was telomeric to but immediately adjacent to the hotspot; in this case, the GCRs recovered were all hotspot-mediated foldback inversions (12 of 12 GCRs) (*Figure 4*; *Supplementary file 3*).

Analysis of potential hairpin-forming sequences centromeric to the induced DSBs suggested that foldback inversions observed tended to use hairpins within 50 bp of the DSB. Within this 50 bp region, potential hairpins used in foldback inversions tended to have shorter loop lengths and longer stem lengths than those that were not used (*Figure 4—figure supplement 1*). Taken together, these data suggest the hypothesis that: (1) if a DSB occurs adjacent (<50 bp) to a short potential hairpin forming sequence, 5' to 3' resection can expose a 3' ssDNA end that can mediate formation of a short hairpin that is likely unstable and must be stabilized by priming DNA synthesis leading to a foldback inversion; and (2) if the DSB is not adjacent to a hairpin forming site, more extensive resection occurs to the hotspot, which forms a much more stable ssDNA hairpin intermediate due to its longer stem length allowing for flap processing followed by priming DNA synthesis leading to a hotspot-mediated foldback inversion (*Figure 2*).

## *SAE2* and *MRE11* specifically suppress hairpins with short loops

We have characterized foldback inversions by WGS both in this study and in our previous studies (*Liang et al., 2018*; *Nene et al., 2018*; *Putnam et al., 2014*; *Srivatsan et al., 2018b*). Comparison of foldback inversions formed in strains with defects in *MRE11* or *SAE2* to those formed in strains with defects in *TEL1*, *MRC1*, and *SWR1* revealed that two distinct classes of ssDNA hairpins mediate foldback inversions. In GCRs isolated from strains with *mre11* or *sae2* defects, the predicted hairpin loop sizes were short (median of 4 and 3 nt, respectively), whereas hairpins in GCRs isolated in strains with wild-type *MRE11* and *SAE2* had larger loops (median 36 nt), which could be many thousands of nucleotides in length (*Figure 5*). This bias was observed even when each unique hairpin site was only counted once to avoid any biases introduced by the inversion hotspot sequence (wild-type *MRE11* and *SAE2*, median 35.5 nt; *mre11* defects, median 5.5 nt; *sae2* defects, median 5.0 nt; *Figure 5*). To distinguish between these two classes of hairpins, here we define 'short-loop' hairpins as being less than 15 nt, which corresponds to 95% of the hairpins and 88% of the hairpin forming sites identified in GCRs formed in strains with *sae2* defects, and 'large-loop' hairpins as being 15 nt or more. For purposes of comparison, the distribution of hairpin loop sizes seen in mutants that give rise to small loop and large loop hairpins, for example *sae2Δ* and *tel1Δ* single mutants, respectively, is provided in the legend to *Figure 5*. In contrast to the effects on hairpin loop size, the length of the hairpin stems observed did not appear to be influenced by the different GCR-inducing mutations tested (*Figure 5*).

In a previous study, we found that the 10 foldback inversions selected in a *tel1Δ* single mutant (10 of 31 GCRs) were mediated by hairpins that often had large loops (median 26 nt, range 4 nt to 44 nt; *Figure 6A*; *Putnam et al., 2014*). In addition, the inversion hotspot was not observed among the inversion sites that mediated the formation of foldback inversions in the *tel1Δ* mutant (*Figure 3A*; $p=5×10^{-5}$, Fisher's exact test; *Putnam et al., 2014*); this is consistent with Tel1 primarily acting to suppress large-loop hairpins and Mre11-Rad50-Xrs2-Sae2 acting to suppress short-loop hairpins that might form in a *tel1Δ* mutant. Therefore, we determined the structures of 11 GCRs selected in a *sae2Δ tel1Δ* double mutant and found that all these GCRs were foldback inversions mediated by small-loop hairpins (median 3 nt; *Figure 3A*; *Figure 6*), and 8 of these were mediated by the inversion hotspot (*Figure 3A*). Thus, the effect of the *sae2Δ* mutation dominates the GCR spectrum in the *sae2Δ tel1Δ* double mutant. Taken together, these results suggest that: (1) Mre11-Rad50-Xrs2-Sae2 preferentially suppresses foldback inversions mediated by short-loop hairpins, (2) foldback inversions formed in the presence of a functional Mre11-Rad50-Xrs2-Sae2 complex, including those in mutants with increased rates of foldback inversion formation, are primarily due to large-loop hairpins that escape surveillance by this complex, and (3) short-loop hairpins may form at higher frequencies relative to large-loop hairpins, as short-loop hairpins accumulate in mutants like the *sae2Δ tel1Δ* double mutant where both types of hairpins can form.

## Efficient DNA resection promotes foldback inversion formation

The CRISPR/Cas9 cleavage results suggest that in many cases the initial step in the formation of foldback inversions could involve resection from some form of DNA damage, such as a DSB or ssDNA gap to expose a ssDNA region containing the hairpin-forming sequence (*Figure 2*). Exo1 and Sgs1/Dna2 define the two major pathways mediating resection from DSBs (*Gravel et al., 2008*; *Mimitou and Symington, 2008*; *Zhu et al., 2008*; therefore, we investigated mutations affecting

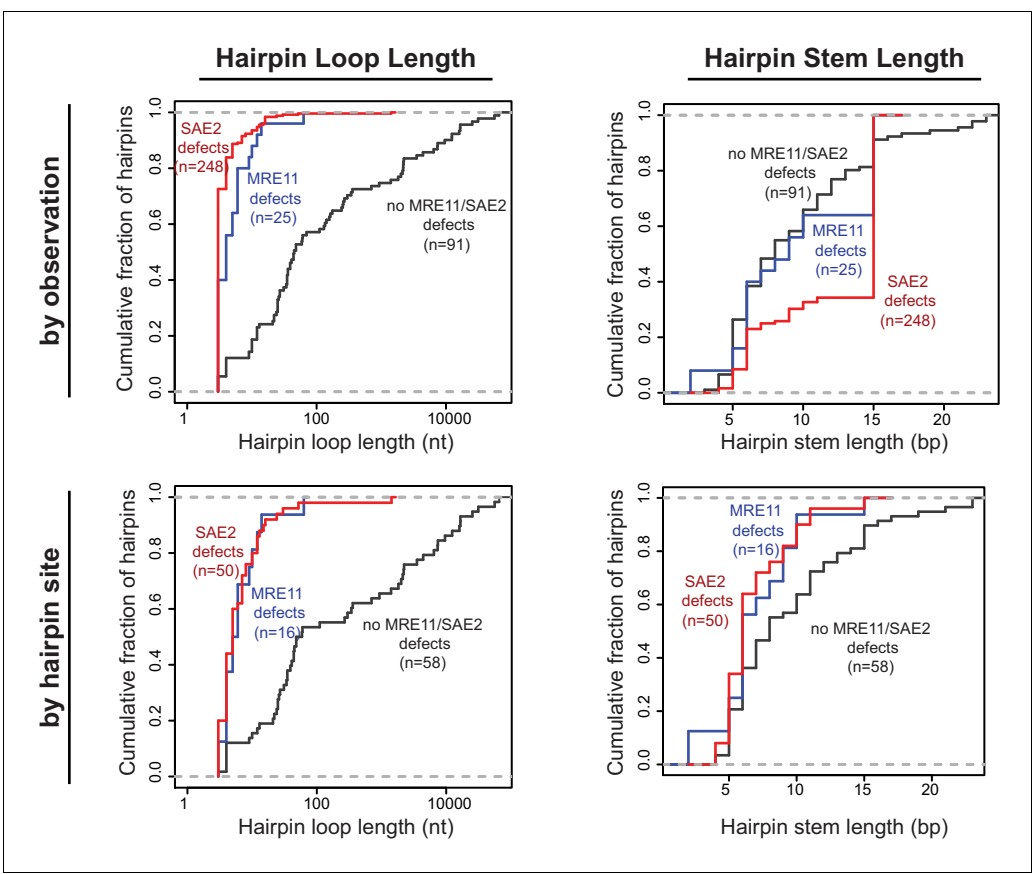

**Figure 5.** Distribution of the loop and stem sizes of the predicted ssDNA hairpins. Cumulative distributions for the loop lengths (left) and hairpin stem lengths (right) for the predicted ssDNA hairpins derived from the inversion junction sequences determined. Hairpins from strains proficient for both *MRE11* and *SAE2* (black), deficient for *MRE11* (blue), and deficient for *SAE2* (red) are plotted separately. Distributions were calculated by counting each observed rearrangement once (top) and by counting each observed inversion site once (bottom). As an example of the difference between the hairpin loop sizes seen in mutants that give rise to small loop and large loop hairpins, the loop size distributions seen in *sae2Δ* and *tel1Δ* mutants are (number of occurrences in parentheses): *sae2Δ* - 3 nt (16), 4 nt (1), 5 nt (1), 8 nt (1), 10 nt (1); and, *tel1Δ* - 4 nt (1), 10 nt (4), 25 nt (1), 35 nt (2), 39 nt (1), 44 nt (1).

The online version of this article includes the following source data for figure 5:

**Source data 1.** Properties of the ssDNA hairpins from observed foldback inversions.

each of these pathways alone and in combination with a *sae2Δ* mutation for their effect on the formation of foldback inversion GCRs.

The *exo1Δ* single mutant primarily accumulated de novo telomere addition GCRs (10 of 11 GCRs; *Figure 7*), whereas the *sae2Δ exo1Δ* double mutant primarily accumulated short loop hairpin-mediated foldback inversion GCRs (16 of 18 GCRs). Both the uGCR and foldback inversion GCR rates of the *sae2Δ exo1Δ* double mutant were ~20% of that of the *sae2Δ* single mutant (*Figure 7A*; *Supplementary file 1*). The foldback inversions selected in the *sae2Δ exo1Δ* double mutant were primarily formed using the inversion hotspot (10 of 16; *Figure 7—figure supplement 1*). We then used CRISPR/Cas9 to induce a DSB ~5 kb telomeric to the inversion hotspot (chrV:25,817–1,754) in the *sae2Δ exo1Δ* double mutant to investigate long-range resection (*Figure 4*). The majority of the DSB-induced GCRs were foldback inversions (9 of 12 GCRs) with an inversion site distribution similar to that observed with the chrV:25,817–1,754 cleavage in the *sae2Δ* single mutant (p=0.3, Fisher's exact test with the categories of DSB-proximal and hotspot inversions). The remaining three DSB-induced GCRs initiated at the DSB-proximal site in the *sae2Δ exo1Δ* double mutant were not foldback inversions (1 each of de novo telomere addition, translocation and interstitial deletion GCRs) and were not observed in the *sae2Δ* single mutant (*Figure 4*). The reduction in GCR rate and the altered GCR

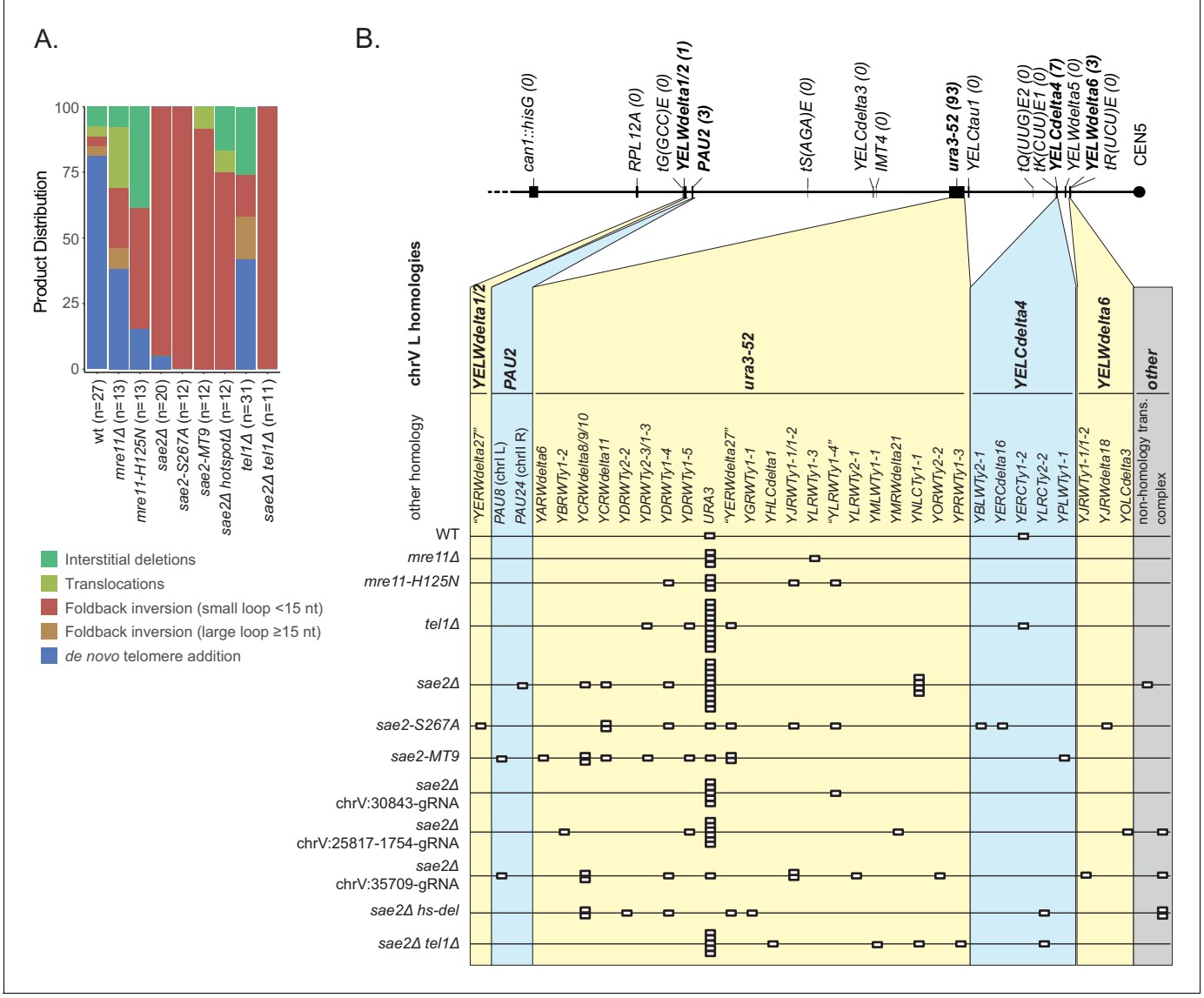

**Figure 6.** Analysis of the GCRs generated in *MRE11*, *SAE2*, and *TEL1* mutants. (**A**) Comparison of the observed GCR spectra. (**B**) Distribution of foldback inversion resolution products observed by genotype. Yellow and blue backgrounds distinguish homologies on chrV L involved in the rearrangement, and columns indicate homologies involved in other regions of the genome. Grey background indicates either non-homology-mediated resolution products or those involving multiple steps, 'complex'.

spectrum in the *sae2Δ exo1Δ* double mutant are consistent with the idea that Exo1 promotes resection to expose hairpin-forming sequences but is not absolutely required for the resection necessary to generate foldback inversions in a *sae2Δ* mutant.

To investigate the effect of disrupting the Sgs1/Dna2 resection pathway, we analyzed viable combinations of the *sgs1Δ*, *yku80Δ*, and *sae2Δ* mutations, as *DNA2* is an essential gene and the synthetic lethality of the *sae2Δ sgs1Δ* double mutant can be suppressed by mutations in *YKU70* or *YKU80* (**Mimitou and Symington, 2010**). Multiple types of GCRs were recovered from the *sgs1Δ* single mutant, and the *sgs1Δ* mutation caused uGCR and foldback inversion GCR rates were modestly increased but had overlapping 95% confidence intervals with the wild-type strain (**Figure 7**; **Supplementary files 1–3**). The *yku80Δ* single mutation reduced the uGCR rate and reduced the rate of de novo telomere addition GCRs by ~30 fold (**Figure 7**; **Supplementary file 1**), consistent with previous observations (**Myung et al., 2001**). In addition, GCRs recovered from the *yku80Δ* single mutant had increased proportions of microhomology-mediated translocations, deletions, and

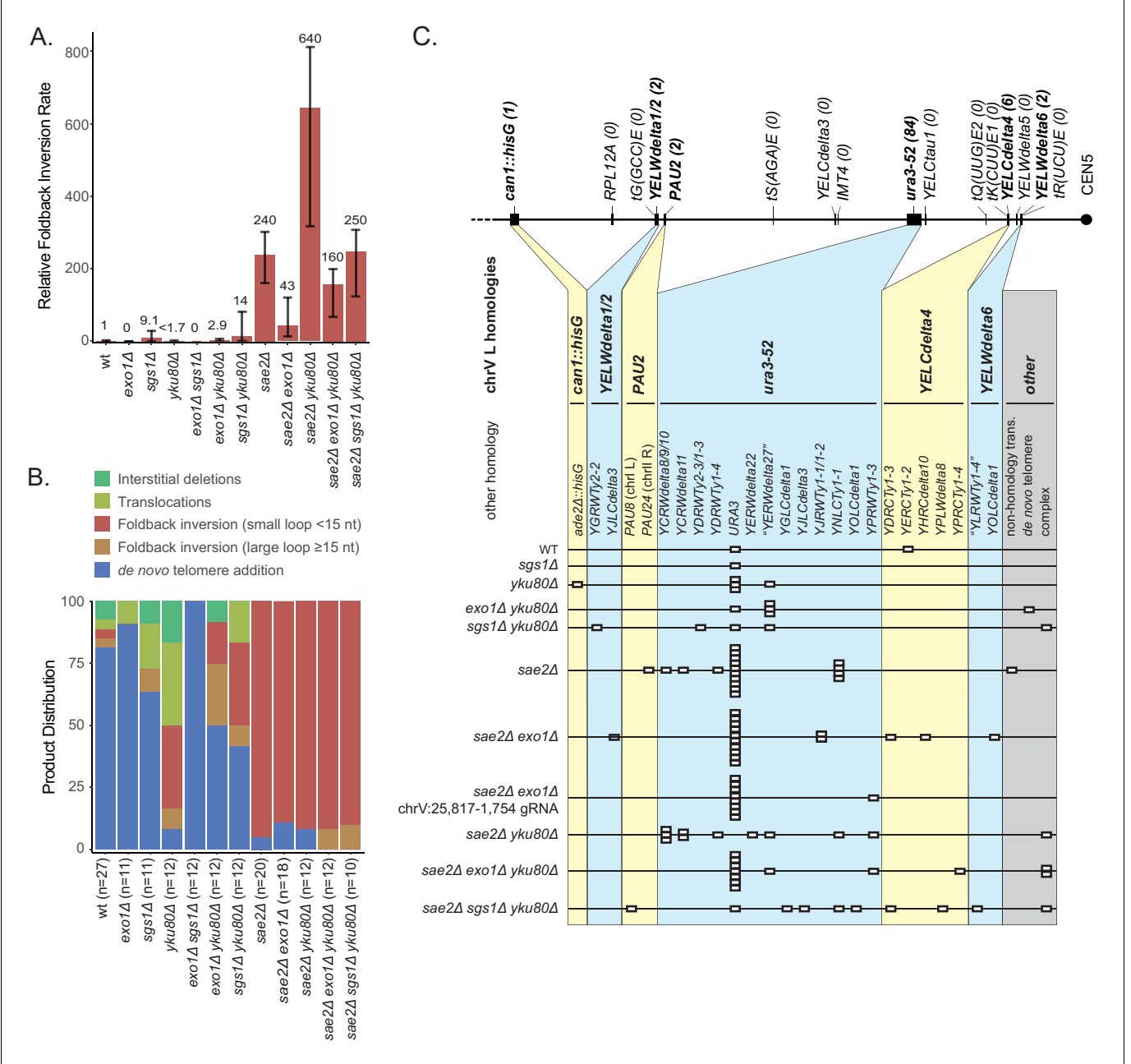

**Figure 7.** Analysis of GCRs generated in strains with mutations in genes that have potential roles in resection. (A) Comparison of the foldback inversion GCR rates relative to wild-type; the foldback inversion rate reported for the *yku80Δ* mutant is the upper bound estimated from the fluctuation results. (B) Comparison of the observed GCR spectra. (C) Distribution of foldback inversion resolution products observed by genotype. Yellow and blue backgrounds distinguish homologies on chrV L involved in the rearrangement, and columns indicate homologies involved in other regions of the genome. Grey background indicates either non-homology mediated resolution products or those involving multiple steps, 'complex'.

The online version of this article includes the following figure supplement(s) for figure 7:

**Figure supplement 1.** Distribution of inversion junctions identified in strains with defects in resection.

foldback inversions mediated by short-loop hairpins (*Figure 7*; *Supplementary file 1*, *Supplementary file 3*). This result suggests that the Yku70-Yku80 complex might play a role in suppressing ssDNA hairpins with short loops. Consistent with this, 30% of the GCRs recovered in the *sgs1Δ yku80Δ* double mutant (4 of 12 GCRs) were foldback inversions mediated by short-loop hairpins compared with none observed in the *sgs1Δ* single mutant strain (0 of 11 GCRs), despite the fact

that the uGCR rates and the foldback inversion rates of the *sgs1Δ* single mutant strain and the *sgs1Δ yku80Δ* double mutant strain were not significantly different. Moreover, the *sae2Δ yku80Δ* double mutant had a threefold increase in the uGCR and foldback inversion GCR rates relative to the *sae2Δ* single mutant (*Figure 7A*), suggesting that suppression of foldback inversions by Yku70-Yku80 and Sae2 are mechanistically distinct; all of the inversion sites used were at the hotspot (*Figure 7—figure supplement 1*). In contrast, the *sae2Δ sgs1Δ yku80Δ* mutant showed a threefold reduction in the uGCR and foldback inversion GCR rates relative to the *sae2Δ yku80Δ* double mutant strain (*Figure 7A*); note that the *sgs1Δ* single and *sgs1Δ yku80Δ* double mutants had similar uGCR rates. Together, these data suggest that Sgs1, like Exo1, also promotes but is not absolutely required for resection leading to foldback inversion formation.

Because the *sae2Δ sgs1Δ* combination was only investigated in the presence of a *yku80Δ* mutation, we tested whether *exo1Δ*, like *sgs1Δ*, would reduce the foldback inversion GCR rate of a *sae2Δ yku80Δ* double mutant strain. Similar to the effect of an *exo1Δ* mutation on the foldback inversion GCR rate of a *sae2Δ* single mutant strain, the *exo1Δ sae2Δ yku80Δ* triple mutant also showed a fourfold reduction in the uGCR and foldback inversion GCR rates relative to the *sae2Δ yku80Δ* double mutant strain (*Figure 7A*; *Supplementary file 1*), and the *exo1Δ* single and *exo1Δ yku80Δ* double mutants had similar uGCR rates. Consistent with the hypothesis that both Exo1 and Sgs1 play redundant roles in mediating resection that promote inverted hairpin formation, the *exo1Δ sgs1Δ* double mutants only accumulated de novo telomere addition GCRs (12 of 12 GCRs; *Figure 7B*), similar to the rearrangements resulting from induced DSBs in *exo1Δ sgs1Δ* double mutants (*Lydeard et al., 2010*).

In aggregate, our results suggest that individual loss of Exo1 or Sgs1 modestly reduces the accumulation of hairpin-mediated GCRs in *sae2Δ* mutants, consistent with a requirement for resection to expose stable hairpin-forming ssDNA sequences and the redundancy of the two pathways in resecting DNA at DSBs, DNA nicks, DNA gaps, and forked DNA structures (*Thangavel et al., 2015*; *Wang et al., 2018*). In addition, our data suggest that Yku70-Yku80 also suppresses the accumulation of short-loop ssDNA hairpins independently of Sae2, as demonstrated by the accumulation of short-hairpin foldback inversion GCRs in the *yku80Δ* single mutant and the *exo1Δ yku80Δ* and *sgs1Δ yku80Δ* double mutants, and the increased foldback inversion GCR rate in the *sae2Δ yku80Δ* double mutant.

## Hairpin intermediates are processed by flap endonucleases

In order to extend a ssDNA hairpin stem by DNA synthesis, the 3′ end of the stem region must be properly base paired, which requires either an appropriately positioned 3′ ssDNA-terminated DSB that can initially prime DNA synthesis or processing of a hairpin with a 3′ ssDNA flap by a flap endonuclease to produce a base paired 3′ end that can prime DNA synthesis (*Figure 2*). To test these possibilities, we first combined a *sae2Δ* mutation with a deletion of the *RAD10* gene, which encodes a subunit of the Rad1-Rad10 3′ flap endonuclease (*Bardwell et al., 1994*). Relative to the wild-type strain, the *rad10Δ* mutation decreased the overall uGCR rate, as previously observed (*Hwang et al., 2005*), and the foldback inversion GCR rate (*Supplementary file 1*). In contrast, the *sae2Δ rad10Δ* double mutant did not have a statistically significant difference in the either the uGCR or the foldback inversion GCR rate compared to that of the *sae2Δ* single mutant (*Figure 8A*; *Supplementary file 1*). Remarkably, the hotspot inversion site was not used as frequently in the *sae2Δ rad10Δ* double mutant as compared to the *sae2Δ* single mutant, and an increased frequency of hairpins with imperfect stems containing mispairs were observed (4 of 11 foldback inversion GCRs) (*Figure 2—source data 1*, *Figure 8—figure supplement 1*).

We next combined the *sae2Δ* mutation with a deletion of the *MUS81* gene, which encodes a subunit of the Mus81-Mms4 endonuclease that cleaves fork structures and 3′ flaps (*Bastin-Shanower and Brill, 2001*; *Ehmsen and Heyer, 2009*; *Fricke et al., 2005*). The *mus81Δ* single mutant did not have a significantly altered uGCR rate compared to wild-type and primarily accumulated de novo telomere addition GCRs (11 of 12 GCRs; *Supplementary file 1*). Similarly, the *sae2Δ mus81Δ* double mutant had a small increase in the uGCR rate and a decrease in the foldback inversion GCR rate compared to the *sae2Δ* single mutant that did not reach statistical significance based on 95% confidence intervals (*Supplementary file 1*). However, the *sae2Δ mus81Δ* double mutant had a decreased proportion of GCRs that were small loop mediated foldback inversions (8 of 16 GCRs) relative to that of a *sae2Δ* single mutant (19 of 20; p=0.005, Fisher's exact test; *Figure 8*;

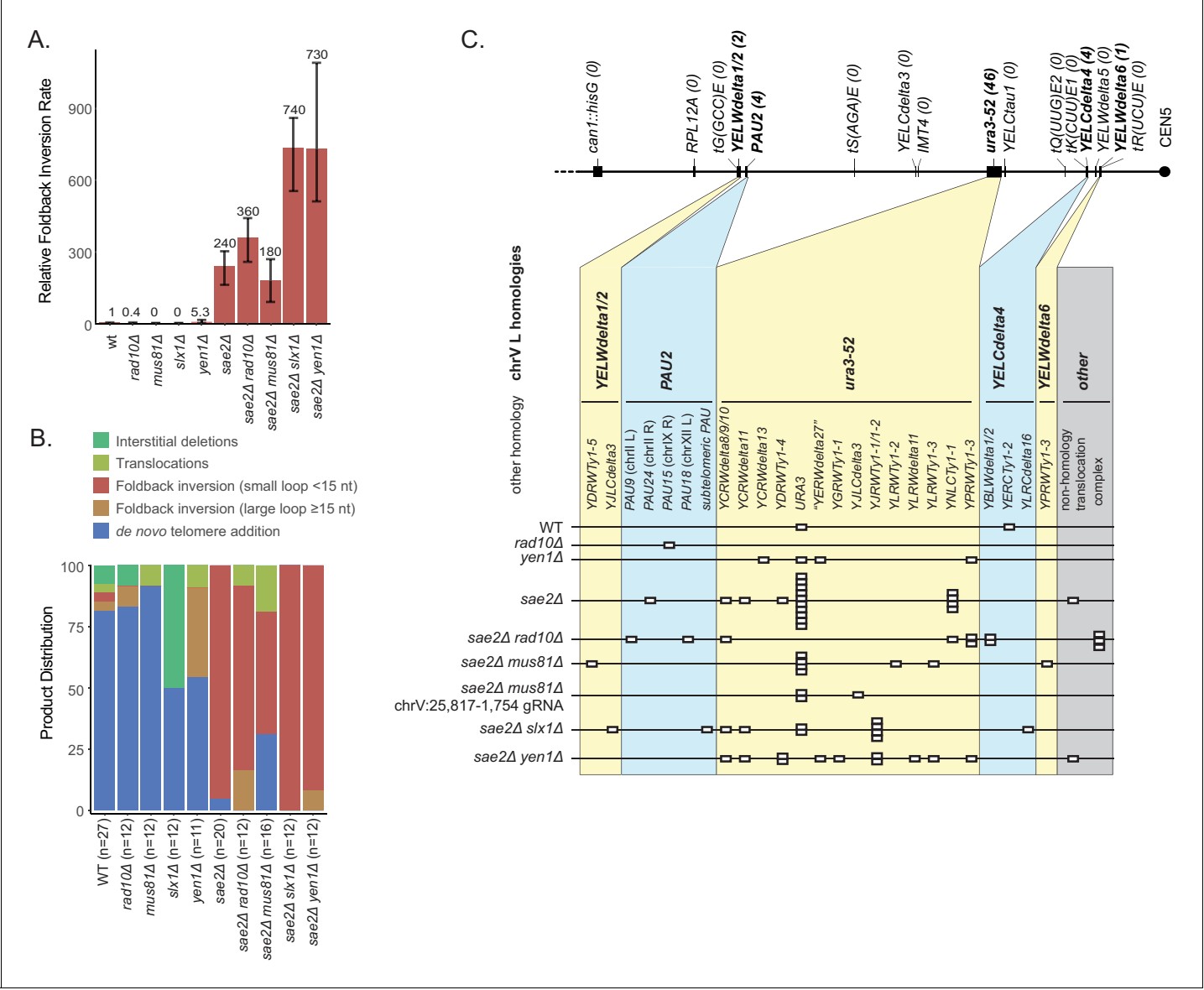

**Figure 8.** Analysis of GCRs generated in strains with mutations in genes encoding flap endonucleases. (**A**) Comparison of the foldback inversion GCR rates relative to wild-type. (**B**) Comparison of the observed GCR spectra. (**C**) Distribution of foldback inversion resolution products observed by genotype. Yellow and blue backgrounds distinguish homologies on chrV L involved in the rearrangement, and columns indicate homologies involved in other regions of the genome. Grey background indicates either non-homology-mediated resolution products or those involving multiple steps, 'complex'.

The online version of this article includes the following figure supplement(s) for figure 8:

**Figure supplement 1.** Distribution of inversion junctions identified in strains with defects in genes encoding flap endonucleases.

*Supplementary file 1*). This change in GCR spectra could occur if Mus81 plays a role in promoting foldback inversions or if possibly the DNA damage arising in a *sae2Δ mus81Δ* double mutant is inefficiently processed into foldback inversion GCRs. To further investigate these possibilities, we used CRISPR/Cas9 to induce a defined DSB telomeric to the inversion hotspot (chrV:25,817–1,754) in the *sae2Δ mus81Δ* double mutant. Only 3 of 12 GCRs recovered were foldback inversions unlike the *sae2Δ* single mutant where all of the GCRs recovered were foldback inversions (11 of 11 GCRs; p=0.0003, Fisher's exact test); in both cases, all of the foldback inversions were mediated by the inversion hotspot (*Figure 4*). The remainder were translocations and interstitial deletions with breakpoints located near the induced DSB. Together the altered spontaneous and DSB-induced spectrum

of GCRs recovered from the *sae2Δ mus81Δ* double mutant relative to that of the *sae2Δ* single mutant is consistent with the possibility that Mus81-Mms4 promotes the formation of foldback inversion GCRs in a *sae2Δ* mutant by processing the 3′ flaps of ssDNA hairpins. These data, however, do not rule out other possibilities in which Mus81-Mms4 promotes foldback inversion formation by acting in steps after the extension of the ssDNA hairpin or by suppressing competing GCR-forming mechanisms.

We then investigated the effect of defects in 5′ flap cleavage on foldback inversion formation, as these flap cleavage endonucleases might act on the loop of the predicted ssDNA hairpin structure. Deletion of *SLX1*, which encodes the catalytic subunit of the Slx1-Slx4 endonuclease (*Fricke and Brill, 2003*), caused a small reduction in the overall uGCR rate and an increased proportion of interstitial deletion GCRs compared to the wild-type strain (*Figure 8*; *Supplementary file 1*). In contrast, the *sae2Δ slx1Δ* double mutant had a threefold increase in both the overall uGCR rate and the rate of accumulating foldback inversion GCRs mediated by small-loop ssDNA hairpins (12 of 12 GCRs) compared to the *sae2Δ* single mutant (19 of 20 GCRs) (p=$7 \times 10^{-6}$, Mann-Whitney U-test; *Figure 8A*; *Supplementary file 1*). Similarly, deletion of *YEN1*, which encodes a 5′ flap/Holliday junction endonuclease (*Ip et al., 2008*), did not cause an increase in the overall uGCR rate but did cause a fivefold increase in the foldback inversion GCR rate compared to the wild-type strain that was of borderline significance (p=0.06, Mann Whitney U-test; *Figure 8A*; *Supplementary file 1*), all of which were mediated by large-loop ssDNA hairpins (4 of 11 GCRs compared to 1 of 27 GCRs for the wild-type strain; p=0.02 Fisher's exact test) (*Figure 8B*), suggesting that *YEN1* may define a large-loop ssDNA hairpin suppression pathway. The *sae2Δ yen1Δ* double mutant had a threefold increase in both the uGCR and foldback inversion GCR rates compared to the *sae2Δ* single mutant; however, the foldback inversions in the *sae2Δ yen1Δ* were primarily mediated by small-loop ssDNA hairpins (11 of 12 GCRs), similar to the *sae2Δ tel1Δ* double mutant. Taken together, these data suggest that 5′ flap cleavage by either the Slx1-Slx4 or Yen1 flap endonucleases suppresses the formation of foldback inversion GCRs mediated by hairpins with short loops in a *sae2Δ* mutant, potentially by cleavage of the loops in ssDNA hairpins that accumulate in the absence of Sae2. The data also suggest that Yen1 plays a role in a pathway that suppresses the accumulation of ssDNA hairpins with large loops.

## Foldback inversions are primarily resolved by homology-mediated secondary rearrangements

The foldback inversion GCRs recovered were monocentric translocations that had almost always undergone additional homology-mediated rearrangements. These foldback inversion resolution products could be divided into three classes.

The first class of resolution products involved a homology-mediated rearrangement between *ura3-52*, located centromeric to the GCR breakpoint region on chrV L, and the oppositely oriented *URA3* sequence, located in the *URA3-CAN1* cassette at the telomeric end of chrV L *Figure 1D*; *Figures 6–9*). This type of secondary rearrangement was previously observed in *tel1Δ* mutants (*Putnam et al., 2014*) and was the most common inverted duplication GCR-associated secondary rearrangement observed (53% of inversion GCRs identified in the *sae2Δ* single mutant and 75% of inversion GCRs identified in the *mre11Δ* single mutant). Remarkably, this *ura3-52/URA3* secondary rearrangement was rarely observed among the GCRs selected in the *sae2* point mutant strains, the GCRs induced by DSBs centromeric to the hotspot (chrV:35,709) in a *sae2Δ* strain, and the spontaneous GCRs observed in a *sae2Δ* strain when the hotspot was also deleted (*Figure 6B*).

The second class of resolution products involved a single rearrangement between a repetitive element between the inversion junction and the chrV centromere and a homology on chrV R or another chromosome (*Figure 1E*; *Figures 6–9*). This class of products has been previously observed (*Liang et al., 2018*; *Nene et al., 2018*; *Pennaneach and Kolodner, 2009*; *Putnam et al., 2014*; *Srivatsan et al., 2018b*).

The third class of resolution products involved complex rearrangements (*Figure 1—source data 1*). These included: (1) multiple homology-mediated rearrangements involving other chromosomes (*Figure 1—source data 1A,C,R,U*); (2) multiple homology-mediated rearrangements restricted to chrV (*Figure 1—source data 1G*); and (3) formation of a second inversion mediated by homologies or ssDNA hairpin intermediates that generated three to six copies of regions of chrV L that were resolved either by additional rearrangements (*Figure 1—source data 1H,I,J,K,M,S*), or duplication of the rest of chrV followed by microhomology-mediated deletion of one of the two resulting copies

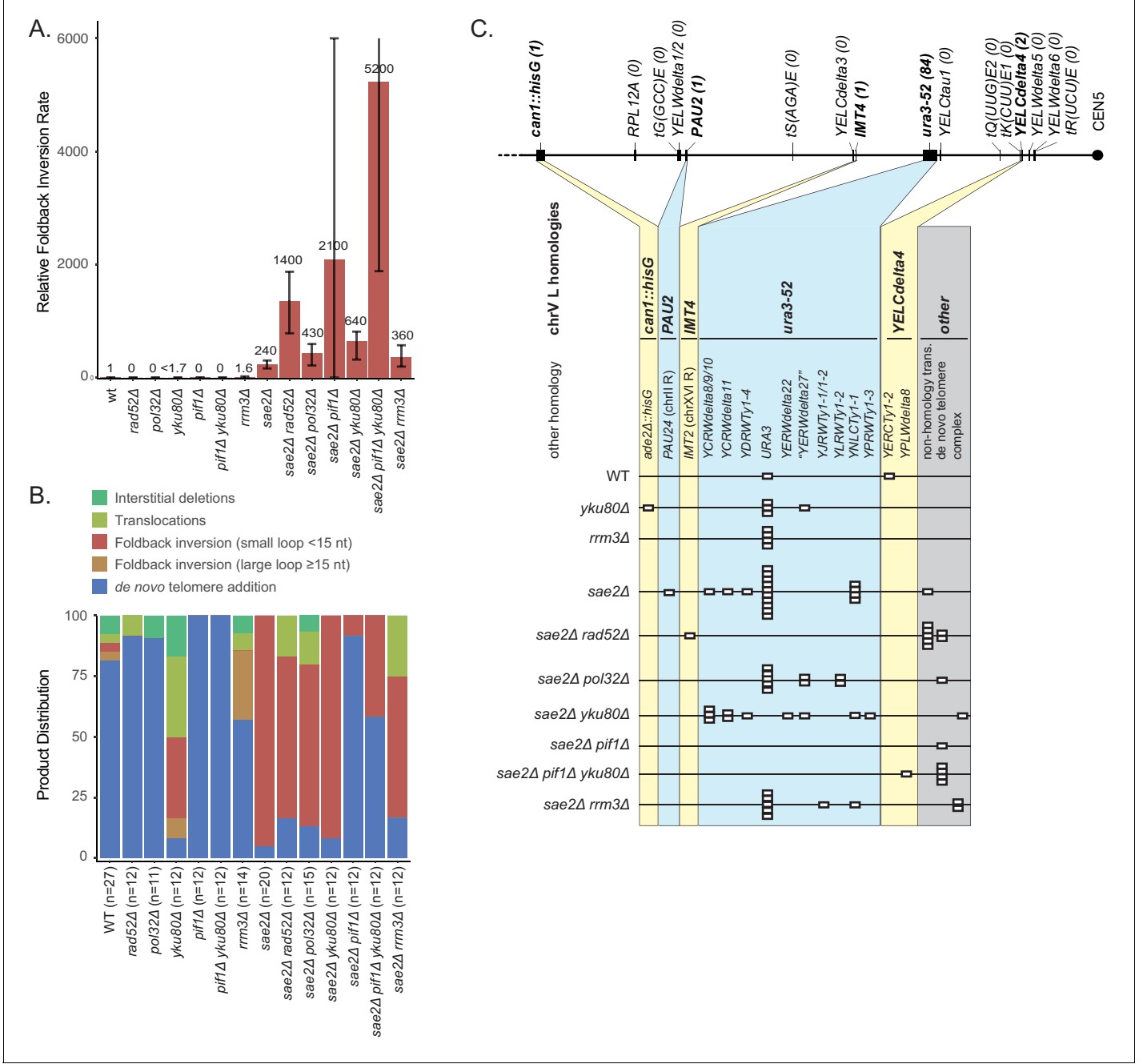

**Figure 9.** Analysis of GCRs generated in strains with mutations in genes that have potential roles in recombination and BIR. (**A**) Comparison of the foldback inversion GCR rates relative to wild-type; the foldback inversion rate reported for the *yku80Δ* mutant is the upper bound estimated from the fluctuation results. (**B**) Comparison of the observed GCR spectra. (**C**) Distribution of foldback inversion resolution products observed by genotype. Yellow and blue backgrounds distinguish homologies on chrV L involved in the rearrangement, and columns indicate homologies involved in other regions of the genome. Grey background indicates either non-homology-mediated resolution products or those involving multiple steps, 'complex'. The online version of this article includes the following figure supplement(s) for figure 9:

**Figure supplement 1.** Distribution of inversion junctions identified in strains with defects in break-induced replication.

of the chrV centromere (*Figure 1—source data 1B,Q*). In addition, foldback inversion GCRs containing variations on the *ura3-52/URA3* secondary rearrangement were observed, including: (1) one GCR in which after forming the *ura3-52/URA3* homology junction, a *HXT13/HXT17* homology-mediated translocation to chrXIV R was observed (*Figure 1—source data 1T*); (2) one GCR in which a

*YELCdelta4/YPRCTy1-2* homology-mediated translocation was formed, followed by formation of a *YPRCTy1-4/ura3-52* junction, and finally formation of a *ura3-52/URA3* junction (*Figure 1—source data 1O*); and (3) one GCR in which a *YELWdelta1/ura3-52* junction was formed, which skipped much of chrV L, followed by resolution by forming the *ura3-52/URA3* rearrangement (*Figure 1—source data 1P*). We also observed a variation on the inversion hotspot in *can1::hisG*, in which a microhomology-mediated translocation between chrV L and chrXV R first occurred, followed by the formation of a hairpin at the equivalent hotspot sequence in *ade2::hisG* on chrXV R, followed by a microhomology-mediated translocation to chrV L and a homology-mediated translocation from chrV L to chrXIV L (*Figure 1—source data 1N*). Some of the multipartite rearrangements observed above could arise either through sequential HR-mediated events or through multi-invasion rearrangements (*Piazza et al., 2017*; *Schmidt et al., 2006*).

## Pol32-dependent break-induced replication does not initiate or resolve foldback inversions

Three possible mechanisms could extend the ssDNA hairpin intermediates and generate the observed GCRs (*Figure 10*). The first mechanism involves strand displacement synthesis followed by template isomerization and homology-mediated recombination mediated by the newly synthesized DNA end. The second mechanism involves establishment of a migrating replication bubble (D-loop), analogous to the one generated during break-induced replication (BIR) (*Donnianni and Symington, 2013*). The third mechanism involves formation of a dicentric chromosome by replication of a hairpin-capped broken chromosome, which is subject to breakage during subsequent rounds of chromosome segregation during cell division. Importantly, BIR could act downstream of all three hairpin-extension mechanisms to mediate the observed homology-mediated secondary rearrangements between the chrV L foldback inversion and targets on chr V R or another chromosome to ultimately capture a telomere-terminated DNA fragment. Moreover, a BIR mechanism is consistent with copying a region from a target chromosome onto the broken chromosome while maintaining an intact target chromosome resulting in the observed duplicated regions. We therefore combined deletions of genes implicated in BIR with a *sae2Δ* mutation to investigate the potential role of BIR or a BIR-like mechanism in either the formation or resolution of foldback inversion GCRs mediated by hairpin formation.

A key step in BIR is Rad52-dependent strand invasion of a single-stranded 3′ end into a homology target to form a D-loop and then prime DNA synthesis. A deletion of *RAD52* caused ~7 fold increase in the uGCR rate when introduced into the wild-type strain, and the GCRs recovered were either de novo telomere addition (11 of 12 GCRs) or translocation GCRs (1 of 12 GCRs) (*Supplementary files 1, 3*). A deletion of *RAD52* also caused an ~8 fold increase in the uGCR rate when introduced into the *sae2Δ* single mutant strain. The *sae2Δ rad52Δ* double mutant had an increased rate of accumulating foldback inversions compared to that of the *sae2Δ* and *rad52Δ* single mutants (*Figure 9*), and the foldback inversions recovered were primarily mediated by the inversion hotspot (8 of 12 GCRs; *Figure 9—figure supplement 1*), indicating that Rad52-dependent HR was not required for forming foldback inversions. Strikingly, the foldback inversions formed in the *sae2Δ rad52Δ* double mutant were not resolved by HR-mediated secondary rearrangements, but were resolved by the formation of microhomology-mediated translocations and de novo telomere addition GCRs (*Figure 9C*; *Supplementary file 3*). Thus, the resolution of the foldback inversions observed in *sae2Δ* single mutants by homology-mediated secondary rearrangements depends on Rad52-dependent HR mechanisms.

The homology-mediated resolution of the foldback inversions generates products that resemble those formed by BIR, which depends upon the DNA polymerase delta subunit Pol32 (*Donnianni and Symington, 2013*; *Donnianni et al., 2019*; *Lydeard et al., 2007*). Deletion of *POL32* did not increase the uGCR rate of a wild-type strain and only increased the uGCR rate of a *sae2Δ* strain by 2.6-fold (*Supplementary file 1*). The *pol32Δ* single mutant GCRs were primarily de novo telomere additions (10 of 11 GCRs), whereas the *sae2Δ pol32Δ* double mutant GCRs were mostly foldback inversions (10 of 15 GCRs; *Figure 9B*; *Supplementary file 3*), which were resolved by homology-mediated secondary rearrangements involving *ura3-52/URA3* (5 of 10 GCRs) and other homology targets (4 of 10 GCRs; *Figure 9C*; *Supplementary file 3*). The spectrum of foldback inversion resolution products formed in the *sae2Δ pol32Δ* double mutant was not statistically different from that of the *sae2Δ* single mutant (p=0.19, Fisher's exact test). This result indicates that the Rad52-dependent

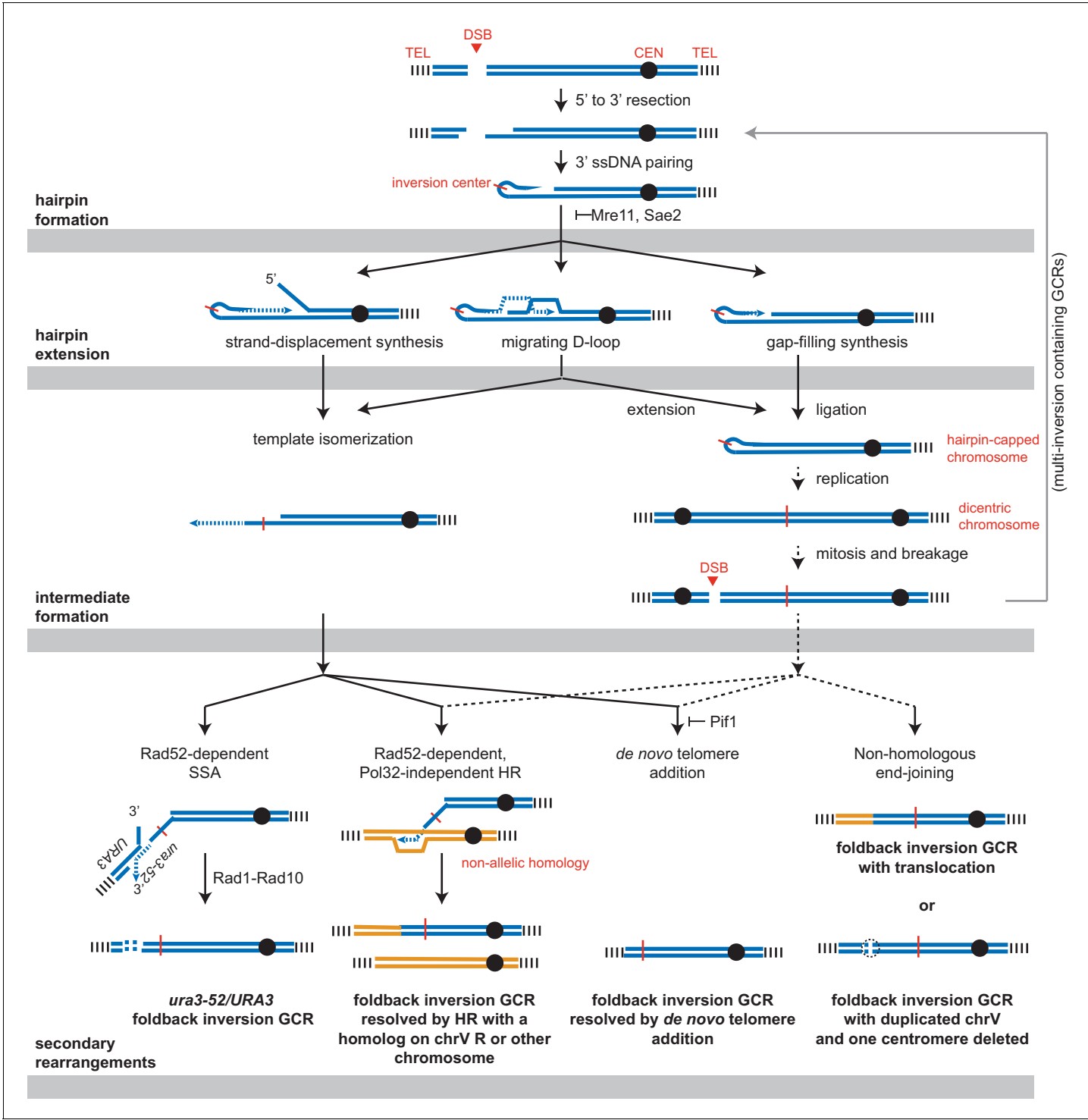

**Figure 10.** Pathways implicated in the formation of foldback inversion GCRs. DNA damage, which is not limited to DSBs, can initiate hairpin formation. Hairpin extension by DNA synthesis could occur by several mechanisms: strand-displacement synthesis, a migrating D-loop, or simple gap-filling synthesis. These mechanisms generate two major types of intermediates: a centromere-containing chromosome fragment with a partially extended and dynamically available 3' ssDNA end or, after DNA replication, a dicentric chromosome V which undergoes breakage during mitosis. The 3' ssDNA end or the broken dicentric chromosome can then participate in multiple types of secondary rearrangements ultimately yielding a monocentric GCR with telomeres at both ends. These rearrangements can be formed in a single step if the resulting GCR is monocentric; however, many of the complex GCRs observed are consistent with multiple rounds of rearrangement and likely involve additional dicentric intermediates (not shown). Most resolution products can be generated from either class of intermediate; however, the *ura3-52/URA3* product, GCRs with multiple foldback inversions, and GCRs that duplicate all of chrV and delete of one of the two centromeres are suggestive of only one of the two proposed intermediates, as indicated.

homology-mediated secondary rearrangements involved in the resolution of foldback inversions formed in the *sae2Δ* single mutant do not depend on a Pol32-dependent BIR mechanism.

The Pif1 DNA helicase is also thought to play a role in BIR because Pif1 acts in vitro to displace the newly synthesized strand and promote D-loop migration-coupled DNA synthesis (*Wilson et al., 2013*). Deletion of *PIF1* caused a 160-fold increase in the uGCR rate relative to the wild-type rate, and the GCRs analyzed were all de novo telomere addition GCRs (*Figure 9A,B*; *Supplementary file 3*), consistent with previous results (*Myung et al., 2001*). The *sae2Δ pif1Δ* double mutant had synergistic increases in the uGCR and foldback inversion GCR rates relative to the single mutants (*Figure 9A*; *Supplementary file 1*). The GCRs isolated from the *sae2Δ pif1Δ* double mutant were primarily de novo telomere addition GCRs (11 of 12 GCRs; *Figure 9B*), whereas the single foldback inversion GCR isolated from the *sae2Δ pif1Δ* double mutant (1 of 12; *Figure 9B*) was resolved by a de novo telomere addition-mediated secondary rearrangement (*Figure 9C*; *Supplementary file 3*). Combining a *yku80Δ* mutation with the *pif1Δ* mutation to reduce the efficiency of de novo telomere addition (*Myung et al., 2001*) only slightly decreased the uGCR rate relative to that of the *pif1Δ* single mutant (*Supplementary file 1*); consistent with this, all the *pif1Δ yku80Δ* double mutant GCRs were de novo telomere addition-mediated GCRs (12 of 12 GCRs) (*Figure 9B*). The *sae2Δ pif1Δ yku80Δ* triple mutant had an increase in the foldback inversion GCR rate of eightfold and threefold relative to the *sae2Δ yku80Δ* and *sae2Δ pif1Δ* double mutants, respectively (*Supplementary file 1*); these foldback inversions were primarily resolved by de novo telomere addition-mediated secondary rearrangements (4 of 5 GCRs; *Figure 9C*). Thus, Pif1 is not required for forming foldback inversions in a *sae2Δ* single mutant but does alter the spectrum of secondary resolution events, likely by suppressing the de novo telomere addition pathway.

The *S. cerevisiae* paralog of Pif1 encoded by *RRM3* has been implicated in promoting telomere replication and sister-chromatid recombination (*Geronimo and Zakian, 2016*; *Muñoz-Galván et al., 2017*). Deletion of *RRM3* did not cause an increase in the uGCR rate or the foldback inversion GCR rate (*Figure 9A*; *Supplementary file 1*). The *sae2Δ rrm3Δ* double mutant had a small increase in the uGCR rate relative to the respective single mutants and an increase in the rate of accumulating microhomology-mediated translocations. Together these observations indicate that the Pif1 homolog Rrm3 has little if any role in the formation or resolution of foldback inversions.

## The *ura3-52/URA3* rearrangement is formed by single-strand annealing

The most common secondary rearrangement seen in foldback inversion GCRs was the *ura3-52/URA3* rearrangement product (*Figure 1D*). As noted above (*Figure 9C*), homology-mediated secondary rearrangements were not observed in the *sae2Δ rad52Δ* mutant strain, consistent with the formation of the *ura3-52/URA3* rearrangement product by some type of HR. The *ura3-52/URA3* resolution product was not observed in the foldback inversions selected in the *rad10Δ* or *sae2Δ rad10Δ* mutant strains (*Figure 8C*). The Rad1-Rad10 nuclease promotes single-strand annealing (SSA) HR by cleaving off non-homologous 3' flaps formed during SSA (*Ivanov and Haber, 1995*; *Prado and Aguilera, 1995*), suggesting that SSA could be involved in the capture of a *URA3*-containing fragment of chrV L. An SSA mechanism would also explain why the captured *URA3*-containing end of chrV was not duplicated (*Figure 1D*), unlike that seen in all other homology-mediated secondary rearrangements in which a telomeric fragment from another chromosome is joined to the broken end of the foldback inversion by non-reciprocal translocation resulting in an intact donor chromosome and duplication of the translocated telomeric fragment. Consistent with an SSA mechanism but not a Pol32-dependent BIR mechanism, the *ura3-52/URA3* product was observed in the *sae2Δ pol32Δ* double mutant (*Figure 9C*; *Supplementary file 3*). In addition, because the ends of both fragments that are joined by SSA must be degraded to expose complementary regions of ssDNA on both fragments, rapid or uncontrolled degradation of the fragments would be expected to reduce the formation of the *ura3-52/URA3* product. Consistent with this idea, the *ura3-52/URA3* secondary rearrangement was also not observed in the foldback inversions selected in the *sae2Δ yku80Δ* double mutant (0 of 11 GCRs) (*Figure 9C*), which lacks the Yku70-Yku80 complex that suppresses extensive resection at DSBs (*Chiruvella et al., 2013*), despite the fact that the rate of accumulating foldback inversion GCRs was threefold higher in the *sae2Δ yku80Δ* double mutant compared to the respective single mutants (*Figure 9A*).

## Discussion

In previous studies, we observed that a large proportion of the GCRs selected in *tel1Δ* mutants were an unusual type of foldback inversion (*Putnam et al., 2014*). These inversions arose by a mechanism whereby a terminal region of chrV L was broken, a region on the centromeric side of the break was duplicated in inverted orientation by a mechanism involving a hairpin junction with a large intervening loop, and the other end of the duplicated region was joined to the telomeric terminal fragment of chrV L, thus capturing a telomere. In the same study, we also observed that *sae2Δ* mutants accumulated GCRs that were consistent with this type of foldback inversion structure and were subsequently shown to have inversion junctions with short intervening sequences between the inverted repeated sequences (*Deng et al., 2015*). In the present study, we have used WGS to determine the complete structure of more than 500 GCRs selected in different mutant strains in order to study the mechanisms by which foldback inversions are formed. The results obtained established a number of key findings. 1) Sae2, the Mre11-Rad50-Xrs2 complex and the Ku complex prevent the formation of foldback inversion GCRs mediated by short-loop hairpins at or near DSBs, and the Slx1-Slx4 flap endonuclease either partially suppresses the foldback inversion GCRs formed in *sae2Δ* single mutants or suppresses DNA damage that can lead to foldback inversions in a *sae2Δ* mutant. 2) Hairpin stems could be as short as 4 bp, and there was a striking inversion hotspot involving formation of a hairpin with a 15 bp stem, but not all possible hairpin-forming sequences mediated the formation of foldback inversion GCRs. 3) Analysis of DSBs induced with CRISPR/Cas9 indicated that while long-range resection from a DSB could target the hairpin-forming hotspot, short-stem hairpin-forming sequences were primarily targeted only if they were adjacent to the DSB. 4) All foldback inversion GCRs underwent secondary rearrangements in which the inverted segment of chrV either captured a telomere-containing chromosome fragment by HR or was broken and then healed by de novo telomere addition; the capture of a terminal fragment of chrV L was mediated by SSA between *ura3-52* and *URA3* in the GCR selection cassette, whereas terminal fragments of other chromosomes were captured by non-reciprocal translocation mechanisms. 5) Pol32-dependent BIR was not required for the resolution of foldback inversions. 6) DSB resection by either Exo1 or Dna2-Sgs1 and hairpin editing by Mus81-Mms4 appeared to play a role in the formation of foldback inversions in *sae2Δ* mutants. Finally, 7) based on data presented here and in our previous studies (*Nene et al., 2018*; *Putnam et al., 2014*; *Srivatsan et al., 2018b*) we were able to identify a second foldback inversion suppressing pathway that prevents the formation of hairpins with large loops and likely involves Yen1, Tel1, Mrc1 and Swr1.

Our analysis of the formation of foldback inversion GCRs in *sae2Δ* strains indicated that the formation of these GCRs is mediated by the formation of ssDNA hairpins. The initiating ssDNA appears to be exposed by resection from the initiating damage by either of the redundant Exo1 and Sgs1/Dna2 pathways resection pathways. Inverted sequences capable of forming the hairpin stems of short loop ssDNA hairpins after resection were located throughout the non-essential breakpoint region probed by the uGCR assay (*Figure 3C*) and throughout the genome. Four bp stem-forming sequences were highly prevalent, but less frequently observed in foldback inversion GCRs, whereas longer stem-forming sequences, including the 15 bp hotspot, were more commonly observed in foldback inversion GCRs. Although precise cleavage can generate a 3′ ssDNA end that can pair with its inverted partner sequence to directly prime DNA synthesis (*Figure 2*), the effects of the *mus81Δ* mutation on the formation of foldback inversion GCRs in the *sae2Δ* mutant and the fact that hairpins can be formed distal to an induced DSB indicate that hairpins with 3′ ssDNA flaps can be formed and require processing by a flap endonuclease (*Figure 2*). In addition, our analysis of CRISPR/Cas9-induced DSBs indicates that short-stemmed hairpins primarily mediate inverted foldback formation when they are proximal to a DSB, whereas long-stemmed hairpins can mediate inverted foldback formation when they are either proximal or distal to the DSB. This could reflect the possibility that short stems are unstable and must be stabilized by priming DNA synthesis, whereas long stems are more stable allowing the formation of flapped structures that can then be processed by 3′ flap endonucleases prior to priming DNA synthesis. Alternatively, the short amount of resection associated with these proximal short-stemmed hairpins (<50 nt) could reflect either (1) inhibitory binding of RPA to long but not short ssDNA resection products which would favor proximal hairpin formation (*Bastin-Shanower and Brill, 2001*) or (2) the small amount of ssDNA DNA that would be exposed by end processing of the DSB by the Mre11-Rad50-Xrs2 complex (*Cannavo and Cejka, 2014*;

*Reginato et al., 2017*; *Wang et al., 2017a*). Interestingly, DSBs induced near short hairpin forming sequences could also induce the formation of other types of GCRs such as translocations, interstitial deletions and DSBs healed by de novo telomere addition indicating that other GCR forming mechanisms can efficiently compete with the formation and fixation of short hairpins.

Two mechanistically distinct types of foldback inversions that can be distinguished by the nature of the initiating ssDNA hairpin have been identified by comparing the GCRs isolated here with those isolated in previous studies (*Nene et al., 2018*; *Putnam et al., 2014*; *Srivatsan et al., 2018b*). Hairpins with small loops (<15 nt) are commonly observed in strains with defects in *MRE11*, *SAE2*, and *YKU80* (*Figures 3* and *5–9*; *Figure 7—figure supplement 1*). This is consistent with previous results indicating that the Sae2-Mre11-Rad50-Xrs2 complex cleaves hairpin structures and the Ku complex protects DSBs from resection, which would both act to suppress the formation of foldback inversions. The fact that foldback inversions dominate the GCR spectrum of *sae2* mutants, but not *mre11* mutants, also suggests that Sae2 is highly specific for hairpin cleavage compared to the Mre11-Rad50-Xrs2 complex but is not absolutely required, as *mre11Δ* mutants have three- to fourfold higher foldback inversion GCR rates than *sae2Δ* mutants. In contrast, strains with defects in *TEL1*, *YEN1*, *SWR1*, and *MRC1* accumulate foldback inversions mediated by hairpins with large loops that can be up to thousands of nucleotides in length (*Figures 5–9*; *Figure 7—figure supplement 1*; (*Nene et al., 2018*; *Putnam et al., 2014*; *Srivatsan et al., 2018b*). These large-loop hairpins appear to avoid surveillance by the activity of Sae2-Mre11-Rad50-Xrs2. In the case of *yen1Δ* mutants, formation of large loops may be due to loss of cleavage of the hairpin loop structure at a site 5′ to the double-stranded stem, which would be consistent with the 5′ flap endonuclease activity of Yen1 (*Ip et al., 2008*). Moreover, the fact that foldback inversions in *sae2Δ yen1Δ* and *sae2Δ tel1Δ* double mutants are dominated by those mediated by small-loop hairpins may suggest that small-loop hairpins form more frequently than large-loop hairpins. This may also explain why *yku80* mutant strains, which have reduced protection of DSBs (*Chiruvella et al., 2013*), accumulate small-loop hairpins.

We envision three possible mechanisms that could extend the ssDNA hairpin intermediates: (1) replication of a hairpin-capped chromosome to generate a dicentric chromosome that subsequently breaks and undergoes secondary rearrangements, (2) resolution of centromere-oriented strand displacement synthesis products, potentially by template switching to a homologous target, and (3) hairpin-primed D-loop formation followed by template switching to a homology target or synthesis of a dicentric chromosome (*Figure 10*). If the role of Pol32 in BIR is to promote D-loop progression via strand-displacement synthesis by DNA polymerase delta (*Stith et al., 2008*; *Wilson et al., 2013*), then the fact that foldback inversion GCRs identified in the *sae2Δ* single mutant and the *sae2Δ pol32Δ* double mutant were indistinguishable suggests that either migrating D-loops do not play a role in extending hairpins or that there are other pathways that are redundant with extension of hairpins by Pol32-dependent progression of migrating D-loops.

The most common secondary rearrangement seen in foldback inversion GCRs, which is mediated by the *ura3-52/URA3* homology (*Figure 1D*), is most easily explained by SSA between a resected *TEL05-URA3* fragment and the single-stranded 3′ end generated by extension of the hairpin to at least the *ura3-52* locus combined with partial or complete dissociation of this 3′ end from the chromosome V template (*Figure 10*). The *TEL05-URA3* fragment would be formed by the same initiating DSB in the same cell cycle as the hairpin-containing centromeric fragment. In contrast, mechanisms involving breakage of a dicentric intermediate chromosome seem less likely, because the *TEL05-URA3*-cassette fragment formed by the initiating DSB would have to be captured by a DSB generated by breakage of a dicentric chromosome V that would occur after cell division; this would allow the *TEL05-URA3*-cassette fragment to segregate away from the broken dicentric chromosome reducing the likelihood of *ura3-52/URA3* rearrangements.

Other secondary rearrangements involve duplication of the chromosomal region targeted by the chrV L homology (typically other chromosomes, but also including chrV R). These secondary rearrangements depended upon Rad52 and could arise from a dissociated single-stranded 3′ end generated by extension of the hairpin or by breakage and resection of a dicentric chromosome intermediate (*Figure 10*). Despite the fact that these products resemble BIR products, their formation was Pol32-independent. This is surprising given the view that Pol32 is absolutely required for DSB-induced BIR (*Lydeard et al., 2007*), but consistent with previous observations that GCRs that resembled BIR-like products selected in a duplication-mediated GCR assay were formed in a *pol32Δ* mutant (*Putnam et al., 2009*).

A final group of secondary rearrangements had structures that are most easily explained by mechanisms involving the resolution of dicentric chromosomes. For example, GCRs involving multiple inversion junctions (in which the most telomeric inversion junction is often used multiple times, for example PGSP4587, PGSP4700, PGSP4771, PGSP4796, PGSP4884, PGSP4967, PGSP5009; *Figure 1—source data 1*) are consistent with an initial product being a broken dicentric chromosome that mediates new ssDNA hairpins from the resulting DSBs; the presence of both hairpins indicates that the DSB was formed such that the original inversion junction in the dicentric was retained. Similarly, GCRs in which all of chromosome V is duplicated, except for an interstitial deletion spanning one of the two copies of the chromosome V centromere are also likely derived from a dicentric intermediate (e.g. PGSP4587, PGSP4967; *Figure 1—source data 1*). Focal deletion of the second centromeres has been previously observed (*Jäger and Philippsen, 1989*; *Kramer et al., 1994*; *Mann and Davis, 1983*; *Pennaneach and Kolodner, 2009*), which may arise from clustering of breaks around centromeres in dicentrics as inferred the pattern of mitotic recombination crossover events (*Song et al., 2013*).

DNA damage is typically repaired using conservative mechanisms that prevent the accumulation of mutations and GCRs. Conservative repair requires the action of both a conservative repair pathway like HR or a damage reversal pathway and pathways that prevent damaged DNAs from being acted on by GCR-promoting pathways (*Figure 11*). In the case of conservative repair by HR, the 3′ overhang used to initiate HR with an allelic site must be protected from competing DNA processing pathways. Our work has demonstrated that at least three pathways can sequester the 3′ overhang and promote the formation of GCRs: de novo telomere addition by telomerase (*Chen and Kolodner, 1999*; *Myung et al., 2001*), foldback inversion formation mediated by small-loop hairpins, and foldback inversion formation mediated by large-loop hairpins (*Figure 11*). Importantly, wild-type cells contain gene products that suppress each of these GCR-promoting reactions. The importance of the latter two GCR-suppression mechanisms is emphasized by the fact that foldback inversions

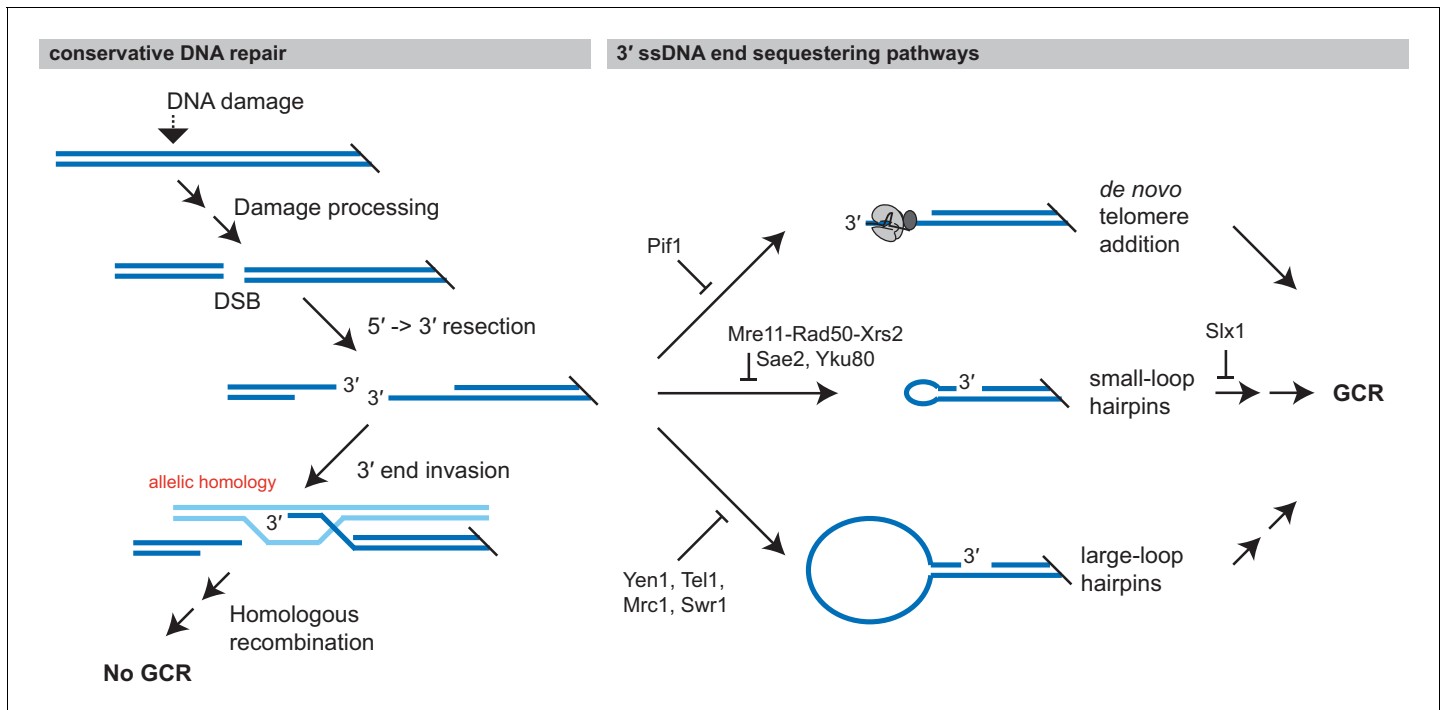

**Figure 11.** Formation of GCRs is promoted by sequestering 3′ ssDNA ends. Conservative repair of many types of DNA damage involves processing the DNA damage to yield a DSB, which is then resected and undergoes allelic HR with the sister chromatid. Allelic HR requires that the 3′ ssDNA ends of the DSB are available to initiate HR. GCRs form when intermediates in conservative repair are acted on by competing DNA processing pathways, particularly if these pathways sequester the 3′ ssDNA end from allelic HR. Analysis of the structure of GCRs, including the analysis presented here has identified three major 3′ ssDNA sequestering pathways, which are suppressed by distinct gene products: de novo telomere addition, formation of small-loop ssDNA hairpins, and formation of large-loop ssDNA hairpins.

have been observed in pancreatic, ovarian, breast, and esophageal cancers (*Campbell et al., 2010*; *Cheng et al., 2016*; *Nik-Zainal et al., 2016*; *Wang et al., 2017b*; *Zhang et al., 2018*) and by the fact that ovarian tumors with increased levels foldback inversions are associated with inferior survival and foldback inversion-mediated high level amplification of oncogenes and focal deletions of tumor suppressors (*Bignell et al., 2007*; *Campbell et al., 2010*; *Wang et al., 2017b*; *Zhang et al., 2018*). Moreover, the computationally determined structural variation signature for these cancer-associated foldback inversions contains clustered inverted duplications and deletions (*Funnell et al., 2019*), which are consistent with the structure of the GCRs observed here.

## Materials and methods

**Key resources table**

| Reagent type (species) or resource | Designation | Source or reference | Identifiers | Additional information |
|---|---|---|---|---|
| Strain, strain background (*S. cerevisiae*) | RDKY6677 | PMID:19641493 | uGCR wild-type | Dr. Richard Kolodner (Ludwig Institute for Cancer Research) |
| Strain, strain background (*S. cerevisiae*) | RDKY6729 | PMID:19641493 | uGCR exo1Δ::HIS3 | Dr. Richard Kolodner (Ludwig Institute for Cancer Research) |
| Strain, strain background (*S. cerevisiae*) | RDKY8032 | PMID:24699249 | uGCR exo1Δ::TRP1 sgs1Δ::HIS3 | Dr. Richard Kolodner (Ludwig Institute for Cancer Research) |
| Strain, strain background (*S. cerevisiae*) | RDKY9771 | This study | uGCR exo1Δ::HIS3 yku80Δ::kanMX4 | Dr. Richard Kolodner (Ludwig Institute for Cancer Research) |
| Strain, strain background (*S. cerevisiae*) | RDKY6686 | PMID:19641493 | uGCR mre11Δ::HIS3 | Dr. Richard Kolodner (Ludwig Institute for Cancer Research) |
| Strain, strain background (*S. cerevisiae*) | HZY2771 | PMID:29505562 | uGCR cir0 mre11-H125N::TRP1 | Dr. Huilin Zhou (Ludwig Institute for Cancer Research) |
| Strain, strain background (*S. cerevisiae*) | RDKY6731 | PMID:19641493 | uGCR mus81Δ::HIS3 | Dr. Richard Kolodner (Ludwig Institute for Cancer Research) |
| Strain, strain background (*S. cerevisiae*) | RDKY6894 | PMID:24699249 | uGCR pif1Δ::HIS3 | Dr. Richard Kolodner (Ludwig Institute for Cancer Research) |
| Strain, strain background (*S. cerevisiae*) | RDKY9773 | This study | uGCR pif1Δ::HIS3 yku80Δ::kanMX4 | Dr. Richard Kolodner (Ludwig Institute for Cancer Research) |
| Strain, strain background (*S. cerevisiae*) | RDKY6703 | PMID:19641493 | uGCR pol32Δ::TRP1 | Dr. Richard Kolodner (Ludwig Institute for Cancer Research) |
| Strain, strain background (*S. cerevisiae*) | RDKY6734 | PMID:19641493 | uGCR rad10Δ::HIS3 | Dr. Richard Kolodner (Ludwig Institute for Cancer Research) |
| Strain, strain background (*S. cerevisiae*) | RDKY6691 | PMID:19641493 | uGCR rad52Δ::HIS3 | Dr. Richard Kolodner (Ludwig Institute for Cancer Research) |
| Strain, strain background (*S. cerevisiae*) | RDKY6735 | PMID:19641493 | uGCR rrm3Δ::HIS3 | Dr. Richard Kolodner (Ludwig Institute for Cancer Research) |
| Strain, strain background (*S. cerevisiae*) | RDKY6687 | PMID:19641493 | uGCR sgs1Δ::HIS3 | Dr. Richard Kolodner (Ludwig Institute for Cancer Research) |
| Strain, strain background (*S. cerevisiae*) | RDKY9445 | This study | uGCR sgs1Δ::HIS3 yku80Δ::kanMX4 | Dr. Richard Kolodner (Ludwig Institute for Cancer Research) |

*Continued on next page*

*Continued*

| Reagent type (species) or resource | Designation | Source or reference | Identifiers | Additional information |
|---|---|---|---|---|
| Strain, strain background (*S. cerevisiae*) | RDKY6738 | PMID:19641493 | uGCR *slx1Δ::kanMX4* | Dr. Richard Kolodner (Ludwig Institute for Cancer Research) |
| Strain, strain background (*S. cerevisiae*) | RDKY6761 | PMID:19641493 | uGCR *tel1Δ::HIS3* | Dr. Richard Kolodner (Ludwig Institute for Cancer Research) |
| Strain, strain background (*S. cerevisiae*) | RDKY9506 | This study | uGCR *yen1Δ::HIS3* | Dr. Richard Kolodner (Ludwig Institute for Cancer Research) |
| Strain, strain background (*S. cerevisiae*) | RDKY8006 | PMID:24699249 | uGCR *yku80Δ::HIS3* | Dr. Richard Kolodner (Ludwig Institute for Cancer Research) |
| Strain, strain background (*S. cerevisiae*) | RDKY6737 | PMID:24699249 | uGCR *sae2Δ::TRP1* | Dr. Richard Kolodner (Ludwig Institute for Cancer Research) |
| Strain, strain background (*S. cerevisiae*) | RDKY9734 | This study | uGCR *sae2Δ::TRP1 hotspotΔ* | Dr. Richard Kolodner (Ludwig Institute for Cancer Research) |
| Strain, strain background (*S. cerevisiae*) | RDKY9472 | This study | uGCR *sae2-S267A* | Dr. Richard Kolodner (Ludwig Institute for Cancer Research) |
| Strain, strain background (*S. cerevisiae*) | RDKY9496 | This study | uGCR *sae2-MT9* | Dr. Richard Kolodner (Ludwig Institute for Cancer Research) |
| Strain, strain background (*S. cerevisiae*) | RDKY8020 | PMID:24699249 | uGCR *sae2Δ::TRP1 exo1Δ::HIS3* | Dr. Richard Kolodner (Ludwig Institute for Cancer Research) |
| Strain, strain background (*S. cerevisiae*) | RDKY9777 | This study | uGCR *sae2Δ::TRP1 exo1Δ::HIS3 yku80Δ:kanMX4* | Dr. Richard Kolodner (Ludwig Institute for Cancer Research) |
| Strain, strain background (*S. cerevisiae*) | RDKY9392 | This study | uGCR *sae2Δ::kanMX4 mus81Δ::HIS3* | Dr. Richard Kolodner (Ludwig Institute for Cancer Research) |
| Strain, strain background (*S. cerevisiae*) | RDKY9447 | This study | uGCR *sae2Δ::TRP1 pif1Δ::HIS3* | Dr. Richard Kolodner (Ludwig Institute for Cancer Research) |
| Strain, strain background (*S. cerevisiae*) | RDKY9779 | This study | uGCR *sae2Δ::TRP1 pif1Δ:: HIS3 yku80Δ::kanMX4* | Richard Kolodner (Ludwig Institute for Cancer Research) |
| Strain, strain background (*S. cerevisiae*) | RDKY9390 | This study | uGCR *sae2Δ::kanMX4 pol32Δ::HIS3* | Richard Kolodner (Ludwig Institute for Cancer Research) |
| Strain, strain background (*S. cerevisiae*) | RDKY9123 | This study | uGCR *sae2Δ::TRP1 rad10Δ::HIS3* | Richard Kolodner (Ludwig Institute for Cancer Research) |
| Strain, strain background (*S. cerevisiae*) | RDKY9504 | This study | uGCR *sae2Δ::TRP1 rad52Δ::HIS3* | Richard Kolodner (Ludwig Institute for Cancer Research) |
| Strain, strain background (*S. cerevisiae*) | RDKY9125 | This study | uGCR *sae2Δ::TRP1 rrm3Δ::HIS3* | Richard Kolodner (Ludwig Institute for Cancer Research) |
| Strain, strain background (*S. cerevisiae*) | RDKY9775 | This study | uGCR *sae2Δ::TRP1 sgs1Δ:: HIS3 yku80Δ::kanMX4* | Richard Kolodner (Ludwig Institute for Cancer Research) |
| Strain, strain background (*S. cerevisiae*) | RDKY9502 | This study | uGCR *sae2Δ::TRP1 slx1Δ::HIS3* | Richard Kolodner (Ludwig Institute for Cancer Research) |

*Continued on next page*

*Continued*

| Reagent type (species) or resource | Designation | Source or reference | Identifiers | Additional information |
|---|---|---|---|---|
| Strain, strain background (*S. cerevisiae*) | RDKY8018 | This study | uGCR *sae2Δ::TRP1 tel1Δ::HIS3* | Richard Kolodner (Ludwig Institute for Cancer Research) |
| Strain, strain background (*S. cerevisiae*) | RDKY9500 | This study | uGCR *sae2Δ::TRP1 yen1Δ::HIS3* | Richard Kolodner (Ludwig Institute for Cancer Research) |
| Strain, strain background (*S. cerevisiae*) | RDKY9443 | This study | uGCR *sae2Δ::TRP1 yku80Δ::kanMX4* | Richard Kolodner (Ludwig Institute for Cancer Research) |
| Recombinant DNA reagent | bRA77 | PMID:28405019 | | Dr. James Haber (Brandeis University) |
| Recombinant DNA reagent | pRS425-Cas9-2XSapI | | | Dr. Bruce Futcher (State University of New York, Stoney Brook) |
| Recombinant DNA reagent | pRS315/sae2-MT9 | PMID:24699249 | | Dr. Richard Kolodner (Ludwig Institute for Cancer Research) |
| Recombinant DNA reagent | pRDK1923 | This study | | Dr. Richard Kolodner (Ludwig Institute for Cancer Research) |
| Recombinant DNA reagent | pRDK1924 | This study | | Dr. Richard Kolodner (Ludwig Institute for Cancer Research) |
| Recombinant DNA reagent | pRDK1929 | This study | | Dr. Richard Kolodner (Ludwig Institute for Cancer Research) |
| Recombinant DNA reagent | pRDK1938 | This study | | Dr. Richard Kolodner (Ludwig Institute for Cancer Research) |
| Recombinant DNA reagent | pRDK1939 | This study | | Dr. Richard Kolodner (Ludwig Institute for Cancer Research) |
| Recombinant DNA reagent | pRDK1940 | This study | | Dr. Richard Kolodner (Ludwig Institute for Cancer Research) |
| Recombinant DNA reagent | pRDK1941 | This study | | Dr. Richard Kolodner (Ludwig Institute for Cancer Research) |
| Sequence-based reagent | pRDK1923-top | This study | | 5′-ATC AAT AGA TCA AAA TCC CCC CC-3′ |
| Sequence-based reagent | pRDK1924-bottom | This study | | 5′-AAC GGG GGG GAT TTT GAT CTA TT-3′ |
| Sequence-based reagent | pRDK1924-top | This study | | 5′-ATC TTG GCT CTG GTC AAT GAT TA-3′ |
| Sequence-based reagent | pRDK1924-bottom | This study | | 5′-AAC TAA TCA TTG ACC AGA GCC AA-3′ |
| Sequence-based reagent | pRDK1924-top | This study | | 5′-ATC TGA ACG CAT GAG AAA GCC CC-3′ |
| Sequence-based reagent | pRDK1294-bottom | This study | | 5′-AAC GGG GCT TTC TCA TGC GTT CA-3′ |
| Sequence-based reagent | pRDK1929-top | This study | | 5′-TCC GTG TTC CAT CCT ACA GAG TTT T-3′ |
| Sequence-based reagent | pRDK1929-bottom | This study | | 5′-TCT GTA GGA TGG AAC ACG GAG ATC A-3′ |

*Continued on next page*

*Continued*

| Reagent type (species) or resource | Designation | Source or reference | Identifiers | Additional information |
|---|---|---|---|---|
| Sequence-based reagent | pRDK1938-top | This study | | 5′-TTA CAT GTT CGA CCG TAC CCG TTT T-3′ |
| Sequence-based reagent | pRDK1938-bottom | This study | | 5′-GGG TAC GGT CGA ACA TGT AAG ATC A-3′ |
| Sequence-based reagent | pRDK1939-top | This study | | 5′-ATA CCT GGA CCC CAG GCA CCG TTT T-3′ |
| Sequence-based reagent | pRDK1939-bottom | This study | | 5′-GGT GCC TGG GGT CCA GGT ATG ATC A-3′ |
| Sequence-based reagent | pRDK1940-top | This study | | 5′-TCA AAT AGG CAT GAT CTT GTG TTT T-3′ |
| Sequence-based reagent | pRDK1940-bottom | This study | | 5′-ACA AGA TCA TGC CTA TTT GAG ATC A-3′ |
| Sequence-based reagent | pRDK1941-top | This study | | 5′-TCT TCC GGG GGC TTT TTT TTG TTT T-3′ |
| Sequence-based reagent | pRDK1941-bottom | This study | | 5′-AAA AAA AAG CCC CCG GAA GAG ATC A-3′ |
| Sequence-based reagent | sae2-S267A repair fragment | This study | | 5′-TGA TAA CTT GAG GAA TAG ATC AAA AGC GCC CCC AGG TTT TGG AAG ACT GGA TTT TCC CTC-3′ |
| Sequence-based reagent | sae2-MT9 amplification forward primer | This study | | 5′-TCC ACC ATT CGA GTC TTG AG-3′ |
| Sequence-based reagent | sae2-MT9 amplification reverse primer | This study | | 5′-TTC CCC TTT CTG CTT TAC CA-3′ |
| Sequence-based reagent | hotspotΔ repair fragment | This study | | 5′-TCA AGA ATT CAG ATC TTC AGT GGT GCA TGA ACG CAT GAG GGC GCG ATA CAG ACC GGT TCA GAC AGG ATA AAG AGG AA-3′ |
| Commercial assay or kit | Gentra Puregene Yeast/Bacteria Kit | Qiagen | 158567 | |
| Commercial assay or kit | TruSeq DNA PCR-free LT kit | Illumina | 15037158 | |
| Software, algorithm | Bowtie 2.2.1 | PMID:22388286 | | http://bowtie-bio.sourceforge.net/bowtie2/index.shtml |
| Software, algorithm | Pyrus 0.7 | PMID:24699249 | | https://sourceforge.net/projects/pyrus-seq/ |

## Plasmid construction

For CRISPR/Cas9-mediated strain construction, complementary 23mer oligonucleotides were annealed and ligated into a *Sap*I-digested pRS425-Cas9-2XSapI. The pRS425-Cas9-2XSapI vector encodes Cas9 and provides a site for cloning and expressing a gRNA encoding sequence and was constructed in Bruce Futcher's laboratory (State University of New York, Stoney Brook). The oligonucleotides 5′-ATC TTG GCT CTG GTC AAT GAT TA-3′ and 5′-AAC TAA TCA TTG ACC AGA GCC AA-3′ were used to generate pRDK1924, which induces DSBs in the *TRP1* gene in a region that is deleted in the *trp1Δ63* allele. The oligonucleotides 5′-ATC AAT AGA TCA AAA TCC CCC CC-3′ and 5′-AAC GGG GGG GAT TTT GAT CTA TT-3′ were used to generate pRDK1923, which induces DSBs in the *SAE2* gene. The oligonucleotides 5′-ATC TGA ACG CAT GAG AAA GCC CC-3′ and 5′-AAC GGG GCT TTC TCA TGC GTT CA-3′ were used to generate pRDK1942, which induces a DSB adjacent to the *can1::hisG* inversion hotspot sequence.

For generating the Gal-inducible CRISPR/Cas9 vectors, complementary oligonucleotides targeting sites on chrV were annealed and ligated into *Bpl*I-digested bRA77, which was a kind gift of Jim

Haber (*Anand et al., 2017*). The oligonucleotides 5′-TCC GTG TTC CAT CCT ACA GAG TTT T-3′ and 5′-TCT GTA GGA TGG AAC ACG GAG ATC A-3′ were used to generate pRDK1929, which cleaves at chrV:34,470. The oligonucleotides 5′-TTA CAT GTT CGA CCG TAC CCG TTT T-3′ and 5′-GGG TAC GGT CGA ACA TGT AAG ATC A-3′ were used to generate pRDK1938, which cleaves at chrV:30,843. The oligonucleotides 5′-ATA CCT GGA CCC CAG GCA CCG TTT T-3′ and 5′-GGT GCC TGG GGT CCA GGT ATG ATC A-3′ were used to generate pRDK1939, which cleaves at chrV:25,817–1,749. The oligonucleotides 5′-TCA AAT AGG CAT GAT CTT GTG TTT T-3′ and 5′-ACA AGA TCA TGC CTA TTT GAG ATC A-3′ were used to generate pRDK1940, which cleaves at chrV:35,709. The oligonucleotides 5′-TCT TCC GGG GGC TTT TTT TTG TTT T-3′ and 5′-AAA AAA AAG CCC CCG GAA GAG ATC A-3′ were used to generate pRDK1941, which cleaves at chrV:34,339–110.

## Strain construction

GCR assays were performed using derivatives of the *S. cerevisiae* strain RDKY6677 (**MATa** *leu2Δ1 his3Δ200 trp1Δ63 lys2ΔBgl hom3-10 ade2::hisG ade8 ura3-52 can1::hisG iYEL072::hphNT1 yel068c:: CAN1/URA3*) (*Putnam et al., 2009*). Standard genetics methods were used to introduce deletion mutations. The *sae2-S267A* mutation was introduced by cutting and subsequent repair of the *SAE2* gene in RDKY6677 with CRISPR/Cas9 by transformation with pRDK1923 and a double-stranded HR repair fragment of which the top strand sequence is 5′-TGA TAA CTT GAG GAA TAG ATC AAA AGC GCC CCC AGG TTT TGG AAG ACT GGA TTT TCC CTC-3′. The *sae2-MT9* mutation was introduced by cutting and subsequent repair of the *TRP1* sequence present in the *sae2::TRP1* disruption cassette in RDKY6737 with CRISPR/Cas9 by transformation with pRDK1924 and a double-stranded PCR product amplified from the pRS313/*sae2-MT9* vector (*Putnam et al., 2014*) with the primers 5′-TCC ACC ATT CGA GTC TTG AG-3′ and 5′-TTC CCC TTT CTG CTT TAC CA-3′. The *sae2Δ hotspotΔ* was generated by CRISPR/Cas9 and HR-mediated repair in the *sae2Δ* strain RDKY6737. Cutting and repair were performed by transformation with pRDK1942 and a double-stranded DNA fragment with the top strand sequence 5′-TCA AGA ATT CAG ATC TTC AGT GGT GC ATG AAC GCA TGA GGG CGC GCG ATA CAG ACC GGT TCA GAC AGG ATA AAG AGG AA-3′, which was generated by annealing oligonucleotides. The strains used in this study are listed in *Supplementary file 7*.

## GCR rate determination

Methods used to determine spontaneous GCR rates and their 95% confidence interval by fluctuation analysis have been described previously (*Srivatsan et al., 2018a*). A single GCR-containing isolate was saved from each culture for sequence analysis. The foldback inversion uGCR rate and the 95% confidence interval for the foldback inversion uGCR rate was calculated by the method described in *Moore et al., 2018*, which propagates the error estimates from the 95% confidence intervals of the uGCR rate and 95% confidence intervals of the proportion of foldback inversions. The 95% confidence interval for the proportion of foldback inversions was calculated by a bootstrap procedure: 100,000 random samples of size *n* were generated from the *n* observations of GCR types in the raw data with replacement (0 = non foldback inversion, 1 = foldback inversion); the proportion of foldbacks in each random sample was determined by dividing the sum of the observations of foldback inversions in the random sample by *n*; and the 95% confidence interval was determined from the 0.025 and 0.975 quantiles of the proportions from all random samples. A pseudo-count of 1 was added to the foldback inversion count or the non-foldback inversion count prior to the bootstrap simulation in cases where either count was zero; this avoids the 95% confidence intervals of 0.0 to 0.0 and 1.0 to 1.0 caused by a lack of diversity in the observed GCRs. This procedure was chosen instead of ones based on asymptomatic approximations, as bootstrap procedures perform better with small sample sizes and completely enumerated finite sample distributions (*Lin et al., 2009*). For a uGCR rate *r* with a 95% confidence interval of $r_{lo}$ to $r_{hi}$, and a foldback inversion proportion *p* with a 95% confidence interval of $p_{lo}$ to $p_{hi}$, the foldback inversion rate *q* with a 95% confidence interval of $q_{lo}$ to $q_{hi}$ was calculated as:

$$q = rp$$

$$q_{lo} = q - q\sqrt{\left(\frac{r - r_{lo}}{r}\right)^2 + \left(\frac{p - p_{lo}}{p}\right)^2}$$

$$q_{hi} = q + q\sqrt{\left(\frac{r - r_{hi}}{r}\right)^2 + \left(\frac{p - p_{hi}}{p}\right)^2}$$

These equations match the procedure described in *Moore et al., 2018*, but the equation in that reference has errors.

### Induction of GCRs with CRISPR/Cas9-induced DSBs

A vector encoding a galactose-inducible CRISPR/Cas9 (pRDK1929, pRDK1938, pRDK1939, pRDK1940, or pRDK1941) was transformed into the appropriate *S. cerevisiae* strain using the conventional lithium acetate method, and transformants were selected on complete synthetic medium lacking leucine (CSM-Leu) plates. The protocol for CRISPR/Cas9 induction and the collection of GCR-containing strains was modified from *Myung, 2003*. Briefly, 2 ml cultures of transformed strains were grown in CSM-Leu media until they reached a density of $2 \times 10^7$ to $4 \times 10^7$ cells/ml. The cells were then washed in sterile distilled water and resuspended in an equal volume of yeast extract-peptone (YP) media containing 2% (w/v) glycerol and 1% succinic acid. After an additional 5 hr of growth, freshly made 50% galactose was added to a final concentration of 2% to induce Cas9 expression, and the cells were grown for an additional 2 hr. After 2 hr of induction, cells were washed with sterile distilled water twice, and resuspended in a $10 \times$ volume of YP media containing 2% glucose (YPD) and grown at 30°C overnight until the culture reached saturation. Cells were then plated onto CSM -Arg plates containing canavanine and 5-fluoroorotic acid to select for GCR-containing clones. A single GCR-containing isolate was saved from each culture for sequence analysis. This protocol resulted in an ~1000 to 10,000-fold induction in the frequency of GCRs, depending on the individual experiment.

### Whole genome paired-end sequencing

Multiplexed paired-end libraries were constructed from 2 μg of genomic DNA purified using the Gentra Puregene Yeast/Bacteria kit (Qiagen). The genomic DNA was sheared using M220 focused-ultrasonicator (Covaris) and libraries were prepared with the TruSeq DNA PCR-free LT kit (Illumina). Pooled libraries were subsequently sequenced on an Illumina HiSeq 4000 using the Illumina GAII sequencing procedure for paired-end short read sequencing.Reads from each read pair were mapped separately by bowtie version 2.2.1 (*Langmead and Salzberg, 2012*) to a reference sequence that contained revision 64 of the *S. cerevisiae* S288c genome (http://www.yeastgenome.org), *hisG* from *Samonella enterica*, and the *hphMX4* marker. Sequence data is available from National Center for Biotechnology Information Sequence Read Archive under accession number PRJNA627970.

### Rearrangement and copy number analysis of paired-end sequencing data

Chromosomal rearrangements were identified after bowtie mapping by version 0.7 of the Pyrus suite (http://www.sourceforge.net/p/pyrus-seq) (*Putnam et al., 2014*). Briefly, read pairs in which both reads uniquely mapped were used to generate the read depth and span depth copy number distributions. The read depth copy number distribution is the number of times each base pair was read in a sample; read depth distributions were the distributions plotted to examine copy number as this distribution is less distorted than the span depth distribution in regions adjacent to repetitive elements. The span depth copy number distribution is the number of times each base pair in a sample was contained in a read or spanned by a pair of reads; span depth distributions were used to statistically distinguish real rearrangements identified by junction-defining discordant read pairs from discordant read pairs that were noise in the data. Read pair data were then analyzed for junction-defining discordant read pairs that indicated the presence of structural rearrangements relative to the reference genome. Associated junction-sequencing reads, which were reads that did not map to the reference but were in read pairs in which one end was adjacent to discordant reads defining a

junction, were used to sequence novel junctions. Most hairpin-generated junctions could be determined using alignments of junction-sequencing reads. For problematic hairpin-generated junctions, the junction sequence could be derived by alignment of all reads in read pairs where one read was present in an 'anchor' region adjacent to the junction of interest and the other read fell within the junction to be sequenced.

## Acknowledgements

This work was supported by NIH grant R01-GM26017 to R.D.K. and the Ludwig Institute for Cancer Research to RDK and CDP.

## Additional information

### Funding

| Funder | Grant reference number | Author |
| --- | --- | --- |
| National Institute of General Medical Sciences | GM26017 | Richard David Kolodner |
| Ludwig Institute for Cancer Research | Lab Funding | Bin-zhong Li<br>Christopher D Putnam<br>Richard David Kolodner |

The funders had no role in study design, data collection and interpretation, or the decision to submit the work for publication.

### Author contributions

Bin-zhong Li, Conceptualization, Investigation, Methodology, Writing - review and editing; Christopher D Putnam, Conceptualization, Data curation, Software, Formal analysis, Supervision, Investigation, Visualization, Methodology, Writing - original draft, Writing - review and editing; Richard David Kolodner, Conceptualization, Formal analysis, Funding acquisition, Visualization, Writing - original draft, Project administration, Writing - review and editing

### Author ORCIDs

Christopher D Putnam https://orcid.org/0000-0002-6145-1265
Richard David Kolodner https://orcid.org/0000-0002-4806-8384

### Decision letter and Author response

Decision letter https://doi.org/10.7554/eLife.58223.sa1
Author response https://doi.org/10.7554/eLife.58223.sa2

## Additional files

### Supplementary files

• Supplementary file 1. Measured uGCR rates, GCR spectra, and calculated foldback inversion rates for strains. These data are displayed in *Figure 1*, *Figure 6*, *Figure 7*, *Figure 8*, and *Figure 9*.

• Supplementary file 2. Statistics for Whole Genome Sequencing results.

• Supplementary file 3. GCR structures.

• Supplementary file 4. De novo telomere addition junction sequences.

• Supplementary file 5. Micro- and non-homology-mediated translocation junction sequences.

• Supplementary file 6. Interstitial deletion junction sequences.

• Supplementary file 7. *Saccharomyces cerevisiae* strains.

• Transparent reporting form

## Data availability

Sequencing data is available from National Center for Biotechnology Information Sequence Read Archive under accession number PRJNA627970. All other data generated are included in the manuscript and supporting files.

The following dataset was generated:

| Author(s) | Year | Dataset title | Dataset URL | Database and Identifier |
|---|---|---|---|---|
| Li B, Putnam C, Kolodner RD | 2020 | *Saccharomyces cerevisiae* foldback inversion GCR sequencing, Apr 21, 2020 | https://www.ncbi.nlm.nih.gov/bioproject/PRJNA627970 | NCBI BioProject, PRJNA627970 |

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
