## [Decision Letter]

**Acceptance summary:**

Nice study trying to define the genetic control of GCRs through WGS and analysis of >500 independent GCRs. General conclusions are that (1) MRX and Ku suppress the short-loop events, (2) Yen and Tel1 suppress the large-loop events, (3) inversions trigger secondary rearrangements, some of which are highly complex, (4) BIR is not a primary contributor to GCR formation and (5) SSA is most often responsible for the capture of a terminal fragment of the broken chromosome. The study determines the nature of the GCRs obtained in MRX and *SAE2* inactivation to focus on a specific type of GCRs mediated by fold-back inversions. Authors propose a pretty coherent mechanism by which such GCRs are generated via a break close to a hairpin followed by resection what determines the initiation of the event leading to the GCR rather than the hairpin becoming a source of breaks. The study supposes a relevant contribution that add novel insight into our understanding of GCR mechanisms.

**Decision letter after peer review:**

Thank you for submitting your article "Mechanisms underlying genome instability mediated by formation of foldback inversions in *Saccharomyces cerevisiae*" for consideration by *eLife*. Your article has been reviewed by three peer reviewers, one of whom is a member of our Board of Reviewing Editors, and the evaluation has been overseen by Kevin Struhl as the Senior Editor The following individual involved in review of your submission has agreed to reveal their identity: Sue Jinks-Robertson (Reviewer #3).

The reviewers have discussed the reviews with one another and the Reviewing Editor has drafted this decision to help you prepare a revised submission.

Summary:

This is a very nice study from the Kolodner lab that finds that GCRs reflect two types of foldback inversions: small loop and large loop. The genetic control of each was determined through a herculean effort that involved the WGS and analysis of >500 independent GCRs. General conclusions are that (1) MRX and Ku suppress the short-loop events, (2) Yen and Tel1 suppress the large-loop events, (3) inversions trigger secondary rearrangements, some of which are highly complex, (4) BIR is not a primary contributor to GCR formation and (5) SSA is most often responsible for the capture of a terminal fragment of the broken chromosome.

The study is able to determine genome wide the nature of the GCRs obtained in MRX and *SAE2* inactivation to focus on a specific type of GCRs mediated by fold-back inversions. Authors propose a pretty coherent mechanism by which such GCRs are generated with a nice and elegant demonstration that it is a break close to a hairpin followed by resection what determines the initiation of the event leading to the GCR rather than the hairpin becoming a source of breaks. This is demonstrated using clever genetic approaches. How the event is processed to end in to the GCR detected is less defined and the most open part of the project that in the future would need to be addressed in a more precise manner. The manuscript is well-written and the full analysis of data are sound. Few point should be addressed, anyhow, before proceeding further with manuscript.

Revisions:

1) A general issue with the rates of uGCR events, especially in comparisons to WT, is that the presentation implies much more accuracy than is inherent in the data on which the graphs in Figure 1 (and Figure 6) are based (Supplementary file 1). Although not explicitly state anywhere (at least that I could find), it is assumed that the error bars on the uGCR rates reflect multiplying the overall GCR rates and Cis by the proportion of the corresponding uGCR events (what I did for many years). This fails to take into account the large errors associated with both small proportions (e.g., 1/14 for WT; 95% CI for 0.071 is 0.0013-0.315; vassarstats.net) as well as proportions based on a small number of events analyzed (4/13 for mre11; 95% CI for 0.31 = 0.13-0.61). A 95% CI can be calculated that takes into account the errors on both the rate and the proportion using the square root of the sum of the squares (see Moore et al., 2018). While this calculation is admittedly over-conservative, especially if one demands that 95% CIs do not overlap, it should be considered as an alternative. With this type of analysis GCR types can be considered significant if the error bars associated with each do not overlap the rate of the other. Combining the errors on the rates and proportions may change some conclusions about genetic control of uGCRs (some significant differences might become insignificant), but will likely not change the major conclusions.

2) It would be better to break Figure 6 into three separate figures – discrete figures that focus on resection, flap/loop processing and secondary rearrangements would make the HUGE amount of data presented much easier to digest.

3) The foldback inversions with secondary rearrangements are confusing when the analysis begins and are in need of a clear accompanying figure. Figure 1D-E are not clear. Figure 6 also does not show the classes of secondary events, although the text implies that it does. A suggestion would be to move Figure 6, figures supplement 3-6 (these were very useful) into the main text. These perhaps could be incorporated into the split versions of Figure 6.

4) Figure 7 is useful, but is doesn't actually show the major classes of secondary events. Is the HR is with the sister, the homolog, a repetitive element or all of the above? SSA should be shown in addition to canonical HR, especially since the last section of the Results is devoted to this. It is really needed a visual to do along with the text.

5) It is unclear that Pol32 data can be used to exclude the third mechanism proposed. It is not a bona-fide BIR mechanism, because it does not require invasion of a free 3' end to initiate DNA synthesis. The 3' end is already sitting on the template DNA, and this may be the critical step at which Pol32 is required in BIR and not just the extension of the DNA synthesis. Also, it cannot be discarded that the displaced flap strand could be used as template to DNA synthesis. Discussion of this model should be revised, including the conclusion about BIR just based on the Pol32 result.

---

## [Author Response]

Revisions:1) A general issue with the rates of uGCR events, especially in comparisons to WT, is that the presentation implies much more accuracy than is inherent in the data on which the graphs in Figure 1 (and Figure 6) are based (Supplementary file 1). Although not explicitly state anywhere (at least that I could find), it is assumed that the error bars on the uGCR rates reflect multiplying the overall GCR rates and Cis by the proportion of the corresponding uGCR events (what I did for many years). This fails to take into account the large errors associated with both small proportions (e.g., 1/14 for WT; 95% CI for 0.071 is 0.0013-0.315; vassarstats.net) as well as proportions based on a small number of events analyzed (4/13 for mre11; 95% CI for 0.31 = 0.13-0.61). A 95% CI can be calculated that takes into account the errors on both the rate and the proportion using the square root of the sum of the squares (see Moore et al., 2018). While this calculation is admittedly over-conservative, especially if one demands that 95% CIs do not overlap, it should be considered as an alternative. With this type of analysis GCR types can be considered significant if the error bars associated with each do not overlap the rate of the other. Combining the errors on the rates and proportions may change some conclusions about genetic control of uGCRs (some significant differences might become insignificant), but will likely not change the major conclusions.

We have adopted the protocol from Moore et al. to propagate errors from the rate and the observed proportion of GCR formation (see new calculations in Supplementary file 1 and Materials and methods). The results of these statistical calculations have been used throughout the paper, and as the reviewers have suspected, the majority of the conclusions have not changed. Our implementation of this method is described in the Materials and methods section.

2) It would be better to break Figure 6 into three separate figures – discrete figures that focus on resection, flap/loop processing and secondary rearrangements would make the HUGE amount of data presented much easier to digest.

As requested by the reviewers, we have split Figure 6 into the new Figures 6-9.

3) The foldback inversions with secondary rearrangements are confusing when the analysis begins and are in need of a clear accompanying figure. Figure 1D-E are not clear. Figure 6 also does not show the classes of secondary events, although the text implies that it does. A suggestion would be to move Figure 6, figures supplement 3-6 (these were very useful) into the main text. These perhaps could be incorporated into the split versions of Figure 6.

As requested by the reviewers, we have added a figure illustrating the classes of homology-mediated secondary rearrangements as the new Figure 10; an improved version of the previous summary figure is now Figure 11. We have additionally added the diagrams illustrating the distribution of secondary events into the new main Figures 6-9.

4) Figure 7 is useful, but is doesn't actually show the major classes of secondary events. Is the HR is with the sister, the homolog, a repetitive element or all of the above? SSA should be shown in addition to canonical HR, especially since the last section of the Results is devoted to this. It is really needed a visual to do along with the text.

We have expanded Figure 7A (new Figure 10) to explicitly show the formation of multiple types of GCRs observed in this manuscript. We agree with the reviewers that this has improved the clarity of the manuscript. We have also ensured that all figures indicating HR specify if the target is allelic or not.

5) It is unclear that Pol32 data can be used to exclude the third mechanism proposed. It is not a bona-fide BIR mechanism, because it does not require invasion of a free 3' end to initiate DNA synthesis. The 3' end is already sitting on the template DNA, and this may be the critical step at which Pol32 is required in BIR and not just the extension of the DNA synthesis. Also, it cannot be discarded that the displaced flap strand could be used as template to DNA synthesis. Discussion of this model should be revised, including the conclusion about BIR just based on the Pol32 result.

We have clarified our discussion of BIR and BIR-like mechanisms in the text, as there are two steps in which BIR-like mechanisms might act. In addition, the new Figure 10 clarifies how BIR might act in the generation of GCRs. In this discussion, we have tried to point out that there could be a distinction between DSB-induced Pol32-dependent BIR studied by others and the formation of the types of products seen here that have structures that could be formed by a BIR type of mechanism; that is to say that we have tried to not use Pol32 dependence as an absolute defining feature of BIR. See P21, L9 to P24, L11, P30, L2-8 and P31, L5-9.

First, the homology-mediated resolution of the foldback inversions typically duplicates the target chromosome from a homology to the telomere. This type of resolution products could be formed by Pol32-dependent BIR. In this case the 3’ end of the foldback chromosome must invade into the homology on the target chromosome. This 3’ end could arise either by breakage and resection of a dicentric intermediate or by displacement of the 3’ end that is extended from the hairpin. These products are formed independently of Pol32, suggesting that a mechanism other than Pol32-dependent BIR is responsible for them.

Second, a conservative replication mechanism involving formation and migration of a D-loop could be involved in the extension of the hairpins shown on the new Figure 10; in this case, the 3’ end is already associated with the template and does not require HR-mediated 3’ strand invasion but rather involves formation of a D-loop later in the process such as by stand invasion/isomerization at the 5' end of the D-loop. As far as we are aware, the mechanistic requirement for Pol32 in BIR has not been established, including role proposed by the reviewers here. The best proposal that we are aware of is the fact that Pol32 is required for extensive strand displacement synthesis (Stith et al., 2008), which is consistent with Pol32 playing a crucial role in migrating one end of the D-loop during BIR. This model suggests that either foldback inversions do not require Pol32-mediated strand displacement synthesis by Pol Δ or that mechanisms that do not involve strand displacement synthesis (such as gap filling and ligation to generate a hairpin-capped chromosome) are efficient enough so that loss of Pol32 does not affect the rates or products observed.